# In vivo imaging of invasive aspergillosis with [18]F-fluorodeoxysorbitol positron emission tomography

Dong-Yeon Kim[1,2,14], Ayoung Pyo [3,4,14], Sehyeon Ji[5], Sung-Hwan You[2,6], Seong Eun Kim[5], Daejin Lim[7], Heejung Kim [8], Kyung-Hwa Lee [9], Se-Jeong Oh[10], Ye-rim Jung[3,6], Uh Jin Kim[5], Subin Jeon[3], Seong Young Kwon[3,6], Sae-Ryung Kang[3,6], Hyang Burm Lee [11], Hoon Hyun[12], So-Young Kim[2,6], Kyung-Sub Moon[10], Sunwoo Lee[13], Seung Ji Kang[5✉] & Jung-Joon Min [2,3,6✉]

Invasive aspergillosis is a critical complication in immunocompromised patients with hematologic malignancies or with viral pneumonia caused by influenza virus or SARS-CoV-2. Although early and accurate diagnosis of invasive aspergillosis can maximize clinical outcomes, current diagnostic methods are time-consuming and poorly sensitive. Here, we assess the ability of 2-deoxy-2-[18]F-fluorosorbitol ([18]F-FDS) positron emission tomography (PET) to specifically and noninvasively detect *Aspergillus* infections. We show that [18]F-FDS PET can be used to visualize *Aspergillus fumigatus* infection of the lungs, brain, and muscles in mouse models. In particular, [18]F-FDS can distinguish pulmonary aspergillosis from *Staphylococcus aureus* infection, both of which induce pulmonary infiltrates in immunocompromised patients. Thus, our results indicate that the combination of [18]F-FDS PET and appropriate clinical information may be useful in the differential diagnosis and localization of invasive aspergillosis.

[1] College of Pharmacy and Research Institute of Pharmaceutical Science, Gyeongsang National University, Jinju, Korea. [2] CNCure Biotech, Hwasun, Korea. [3] Innovation Center for Molecular Probe Development, Department of Nuclear Medicine, Chonnam National University Medical School and Hwasun Hospital, Hwasun, Korea. [4] Accelerator & RI Development Team, Korea Atomic Energy Research Institute, Jeongeup, Korea. [5] Department of Internal Medicine, Chonnam National University Medical School, Hwasun, Korea. [6] Institute for Molecular Imaging and Theranostics, Chonnam National University Medical School, Hwasun, Korea. [7] Division of Biomedical Convergence, College of Biomedical Science, Kangwon National University, Chuncheon, Korea. [8] Korea Radioisotope Center for Pharmaceuticals, Korea Institute of Radiological & Medical Sciences, Seoul, Korea. [9] Department of Pathology, Chonnam National University Medical School, Hwasun, Korea. [10] Department of Neurosurgery, Chonnam National University Medical School, Hwasun, Korea. [11] Department of Agricultural Biological Chemistry, Chonnam National University, Gwangju, Korea. [12] Department of Biomedical Sciences, Chonnam National University Medical School, Hwasun, Korea. [13] Department of Chemistry, Chonnam National University, Gwangju, Korea. [14] These authors contributed equally: Dong-Yeon Kim, Ayoung Pyo. ✉email: sseungi@gmail.com; jjmin@jnu.ac.kr

Fungal infections may be life-threatening in various types of patients, including patients with hematologic malignancies; those who have undergone solid organ or hematopoietic stem cell transplantation; those receiving chemotherapy, immunosuppressive therapy, or prolonged corticosteroid treatment; and those with prolonged stay in the intensive care unit (ICU)[1–3]. *Aspergillus fumigatus* (*A. fumigatus*) is the most frequent cause of invasive fungal disease in humans, usually affecting the lungs, sinuses, and brain[4]. Global models estimate that more than 3,000,000 individuals experience chronic pulmonary aspergillosis and over 250,000 experience invasive aspergillosis (IA) annually, with these infections associated with high morbidity and mortality rates[5]. IA is a frequent complication in immunocompromised patients such as those with neutropenia[2,6], but the incidence of IA in non-neutropenic patients with underlying lung diseases (e.g., chronic obstructive pulmonary disease, asthma, lung cancer, and viral pneumonia)[7,8] has also increased. Several recent reports have described COVID-19-associated pulmonary aspergillosis (CAPA)[9–12], raising concerns that CAPA could worsen outcomes of COVID-19[13].

Early recognition of IA and immediate administration of proper antifungal agents are crucial for patient survival, but early recognition is usually difficult because the pace of progression leaves only a narrow window for diagnosis[14]. Moreover, invasive procedures for definitive diagnosis are not possible due to patient comorbidities, such as coagulopathy and respiratory compromise[14]. Although mycological culture of *Aspergillus* from infected tissues is the reference standard for IA diagnosis, this method is invasive, with low sensitivity and a long turnaround time[7,14]. The detection of circulating antigens such as galactomannan (GM) and β-D-glucan (BDG) is regarded as a rational first step for diagnosis of IA, but this indirect microbiologic test has low sensitivity and specificity[7,14]. Although computed tomography (CT) is frequently used to make decisions regarding the treatment of invasive pulmonary aspergillosis (IPA), radiographic findings in IPA are generally non-specific and differ by neutrophil count[15,16]. In patients with mixed infection such as CAPA, the lack of specific radiologic signs of IPA complicates its radiologic diagnosis[13]. Positron emission tomography (PET) using 2-deoxy-2-[18]F-fluoro-D-glucose ([18]F-FDG), a successful molecular imaging method in oncology, was unable to distinguish IPA from sterile inflammation, cancer, and bacterial infection in both preclinical and clinical studies[17–19]. Other molecular imaging tracers, such as [99m]Tc-labeled phosphorodiamidate morpholino (MORF) oligomer probes[20], [68]Ga- and [89]Zr-labeled siderophores[21–24], and the [64]Cu-DOTA-labeled *Aspergillus*-specific mouse monoclonal antibody (mAb) mJF5[25], have been

developed, each tracer has limitations, including a lack of signal intensification at the site of infection and/or unknown toxicity in humans.

Because the sugar-free sweetener sorbitol is a metabolic substrate for Enterobacterales, the ability of 2-deoxy-2-[18]F-fluorosorbitol ([18]F-FDS) was assessed for its ability to image pathogenic *E. coli*[26], with this method found to be selective for Gram-negative pathogenic bacteria[27,28]. Intriguingly, sorbitol can act as a sole carbon source supporting high fungal growth and sporulation[29]. The gene *mpkC*, which encodes a mitogen-activated protein kinases, was found to play a key role in controlling the growth of *A. fumigatus* in response to sorbitol as a carbon source[30]. These findings suggested that [18]F-FDS has potential as a suitable PET probe to image *A. fumigatus* infections in vivo.

Here, we evaluate the in vitro and in vivo properties of [18]F-FDS as an imaging agent for *A. fumigatus* infection in various mouse models. We show that [18]F-FDS selectively concentrates in regions of *A. fumigatus* infection and differentiates *A. fumigatus* infection from other infectious and non-infectious causes. Our results suggest that this radiotracer can be used as a PET molecular imaging agent to obtain high quality images in the diagnosis of *A. fumigatus* infection.

## Results

**[18]F-FDS preparation and in vitro uptake studies.** The identity and radiochemical purity of [18]F-FDS were evaluated by comparing the radio-TLC $R_f$ value of [18]F-FDS with that of cold [19]F-FDS[31,32]. The total time required to prepare [18]F-FDS was within 30 min, and its overall decay-corrected radiochemical yield was approximately 35–40%. [18]F-FDS had a specific activity greater than $13.8 \pm 1.3$ GBq/μmol and a radiochemical purity greater than 98%.

The in vitro [18]F-FDS uptake by *A. fumigatus*, *Rhizopus arrhizus* (*R. arrhizus*), and *Candida albicans* (*C. albicans*), clinically important fungal pathogens in immunocompromised patients, was assessed, with *Escherichia coli* (*E. coli*) and *Staphylococcus aureus* (*S. aureus*) acting as positive and negative controls, respectively[26]. [18]F-FDS uptake by *A. fumigatus*, *R. arrhizus*, and *C. albicans* was comparable to its uptake by *E. coli*, whereas heat-killed pathogens and the Gram-positive bacterium *S. aureus* did not significantly accumulate the probe (Fig. 1 and Supplementary Fig. 1). We also tried to visualize *A. fumigatus* directly under a confocal fluorescence microscopy using near-infrared fluorescent sorbitol (sorbitol-ZW800-1) and showed that the sorbitol-ZW800-1 was mainly accumulated in germinated *A. fumigatus* (Supplementary Fig. 2).

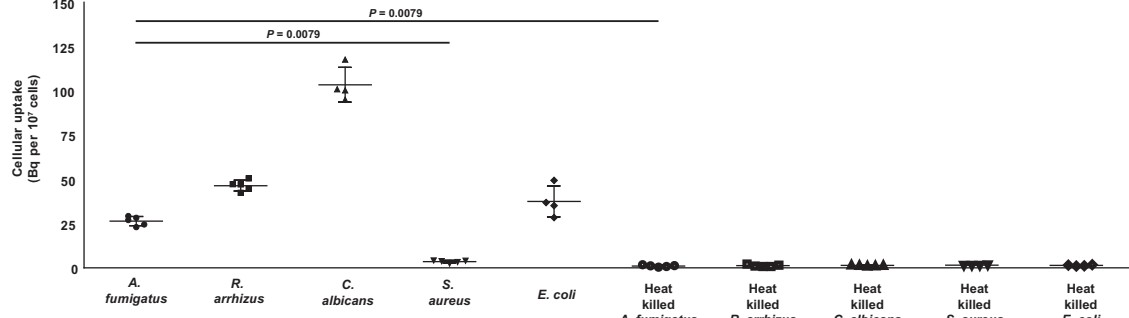

**Fig. 1 Cellular uptake of radioactivity 1 h after treatment with [18]F-FDS of *A. fumigatus*, *R. arrhizus*, *C. albicans*, *S. aureus* (−), and *E. coli* (+) cells and corresponding heat-killed cells.** *E. coli* was the positive control, and *S. aureus* and heat-killed pathogens were negative controls. Data are expressed as the mean of absolute accumulation activity (Bq) ± SD (per $1 \times 10^7$ cells) of four or more replicate experiments (*A. fumigatus*, *R. arrhizus*, *S. aureus*, heat killed *A. fumigatus*, heat killed *R. arrhizus*, and heat killed *C. albicans*; n = 5, *C. albicans*, *E. coli*, heat killed *S. aureus*, and heat killed *E. coli*; n = 4). Statistical significance was calculated using two-tailed Mann–Whitney U tests.

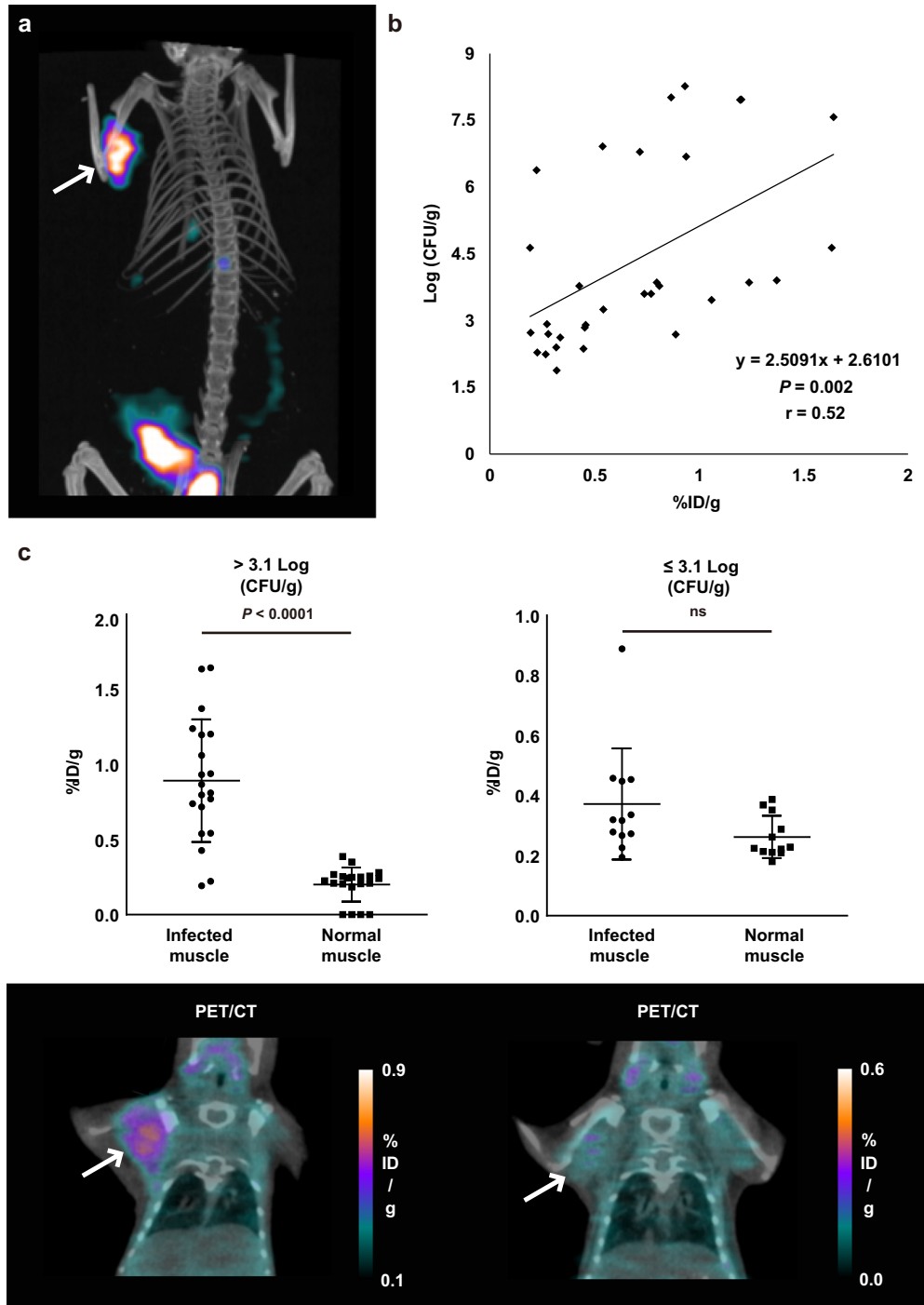

**Fig. 2 MicroPET/CT imaging of $^{18}$F-FDS in mice with *A. fumigatus*-infected myositis 2 h after injection and correlation between $^{18}$F-FDS and number of *A. fumigatus* cells. a** Representative $^{18}$F-FDS PET/CT images in mice with *A. fumigatus*-infected myositis 2 h after *i.v.* injection of $^{18}$F-FDS (white arrow, $n = 9$). **b** Correlation between uptake of $^{18}$F-FDS and number of *A. fumigatus* conidia ($n = 32$). Mice were inoculated in the right shoulder with $1 \times 10^1$ to $1 \times 10^8$ *A. fumigatus* conidia, and microPET/CT imaging was performed 3 days later. The mice were subsequently sacrificed and the CFU were counted immediately. The correlation was found to be y = 2.5091x + 2.6101 ($r = 0.52$, $P = 0.002$). X axis: %ID/g of $^{18}$F-FDS, Y axis: Log (CFU/g) of *A. fumigatus*. **c** $^{18}$F-FDS uptake by infected and normal muscles in each subgroup divided by the cutoff number of *A. fumigatus* conidia (3.1 log CFU/g, ns = not significant, left; $n = 20$, right; $n = 12$). Data are expressed as the mean ± SD. Statistical significance was calculated using two-tailed Mann–Whitney U tests.

**$^{18}$F-FDS can detect and differentiate *A. fumigatus*-infected myositis from sterile inflammation, tumor, and Gram-positive bacterial infection.** We next investigated whether $^{18}$F-FDS PET could detect *A. fumigatus* infection in mouse models. *A. fumigatus*-infected myositis was generated by inoculation of $1 \times 10^8$ of *A. fumigatus* conidia into the right shoulder muscle of

immunosuppressed BALB/c mice. The infected muscle was clearly visualized by $^{18}$F-FDS, with its activity retained for 2 h (Fig. 2a and Supplementary Fig. 3). $^{18}$F-FDS uptake by infected muscle was significantly higher than its uptake by the contralateral normal muscle, with a mean ratio at 2 h of 8.90 ± 1.81 (Supplementary Fig. 4). Rapid clearance and low non-specific

**Table 1 Biodistribution of $^{18}$F-FDS (2 h, 7.4 MBq, $n = 8$) in mice with *A. fumigatus*-infected myositis.**

| Organs | %ID/g $^{(2\ h)}$ |
|---|---|
| Blood | 0.22 ± 0.05 |
| Heart | 0.82 ± 0.14 |
| Lung | 0.97 ± 0.16 |
| Liver | 0.46 ± 0.15 |
| Spleen | 1.59 ± 0.25 |
| Stomach | 0.74 ± 0.42 |
| Intestine | 1.75 ± 0.93 |
| Kidney | 0.79 ± 0.17 |
| Pancreas | 1.42 ± 0.26 |
| Bone | 0.72 ± 0.13 |
| Brain | 0.33 ± 0.04 |
| Skin | 0.51 ± 0.12 |
| Normal muscle | 0.28 ± 0.03 |
| Infected muscle | 1.84 ± 0.38 |

Radioactivity was normalized relative to the weight of tissue and the amount of radioactivity injected.
Data are expressed as mean ± SD % ID/g.

binding in normal organs resulted in high infection-to-lung, and -liver ratios at 2 h of 9.27 ± 3.32 and 9.33 ± 3.73, respectively (Supplementary Fig. 4). Dynamic $^{18}$F-FDS PET imaging (over 120 min) in mice with *A. fumigatus*-infected myositis was also performed (Supplementary Movie). $^{18}$F-FDS PET signals could be seen consistently in the infected site even at 120 min, whereas blood pool activities in the heart and liver dissipated after 60 min. This finding indicated that the $^{18}$F-FDS PET signal in *A. fumigatus* infection is independent of blood pool effects. *A. fumigatus*-infected myositis was confirmed by pathological examination after PET analysis (Supplementary Fig. 5).

The limit of detection of $^{18}$F-FDS in *A. fumigatus*-infected myositis was determined by microPET analysis in mice inoculated with various numbers of *A. fumigatus* conidia (from $1 \times 10^1$ to $1 \times 10^8$, Fig. 2b). $^{18}$F-FDS uptake was semi-quantitatively correlated with the number of *Aspergillus* conidia in mice with infected muscle ($r = 0.52$, $P = 0.002$). Receiver operating characteristic (ROC) curve analysis of the minimum number of *A. fumigatus* conidia that could be visually detected on $^{18}$F-FDS PET revealed that the optimal cutoff number of *A. fumigatus* conidia was 3.1 log (CFU/g). The sensitivity, specificity, positive predictive value, negative predictive value, and accuracy of $^{18}$F-FDS PET in the visual detection of *A. fumigatus*-infected myositis (more than 3.1 log CFU/g of tissue) were 90.0, 83.3, 90.0, 83.3, and 87.5%, respectively. A comparison of $^{18}$F-FDS uptake by infected and normal muscle in each subgroup divided by the cutoff number of *A. fumigatus* conidia found that, above the cutoff level (≥ 3.1 log CFU/g), $^{18}$F-FDS uptake was significantly higher in infected than in normal muscle (0.89 ± 0.41%ID/g vs. 0.20 ± 0.11%ID/g, $P < 0.0001$). Below the cutoff level, however, $^{18}$F-FDS uptake did not differ significantly in infected and normal muscle (0.37 ± 0.18%ID/g vs. 0.26 ± 0.07%ID/g, $P = 0.0531$) (Fig. 2c).

Next, we evaluated whether $^{18}$F-FDS PET could distinguish *Aspergillus* infection from sterile inflammation, tumor, and Gram-positive bacterial infection in vivo. Live *A. fumigatus* was injected into the right shoulders and a 10-fold higher number of heat-killed *A. fumigatus* into the left shoulders of immunosuppressed mice. The right shoulders of other immunosuppressed mice were inoculated with live *S. aureus* or CT26 colon cancer cells. Infection with *S. aureus* and sterile inflammation were confirmed by histopathological analysis. Consistent with in vitro uptake data and the in vivo experiments describe above, $^{18}$F-FDS

readily concentrated in the *Aspergillus*-infected shoulders, but not in the shoulders with sterile inflammation, CT26 tumor, and *S. aureus* infection (Fig. 3a–c), with $^{18}$F-FDS uptake by infected muscle being 3.9-fold higher than by shoulders with sterile inflammation (0.95 ± 0.10%ID/g vs. 0.24 ± 0.01%ID/g, $P = 0.0286$, Fig. 3d). By contrast, reference imaging showed that $^{18}$F-FDG accumulated not only in *Aspergillus*-infected shoulders, but also in sterile inflamed, *S. aureus*-infected, and tumor-engrafted shoulders (Fig. 3a–c). $^{18}$F-FDG uptake by shoulders with sterile inflammation was higher than that by shoulders with *A. fumigatus* infection (8.57 ± 0.77%ID/g vs. 3.63 ± 0.86%ID/g, Fig. 3e).

The in vivo biodistribution study performed 2 h after tracer injection also showed selective concentration of $^{18}$F-FDS in the shoulders with *A. fumigatus*-infected myositis, with concentrations of $^{18}$F-FDS in these shoulders being 8.4-, 1.8-, 4.0-, 5.5-, and 6.6-fold higher than in blood, lungs, liver, brain, and normal muscle, respectively (Table 1).

**$^{18}$F-FDS PET can selectively detect and localize *A. fumigatus* infection in the lung and brain.** We then extended our findings to mouse models of IPA. Immunosuppressed BALB/c mice were intranasally inoculated with $1 \times 10^8$ *A. fumigatus* conidia or *S. aureus*. Pulmonary infection with *A. fumigatus* or *S. aureus* was confirmed by histopathological analysis (Fig. 4b and Supplementary Fig. 6). $^{18}$F-FDS readily concentrated in the infected lung, but not in the normal lung, and $^{18}$F-FDS PET could clearly visualize *Aspergillus* infection in the lung parenchyma (Fig. 4a and Supplementary Fig. 7). More precise examination using high-resolution autoradiography clearly demonstrated the specific uptake of $^{18}$F-FDS by *A. fumigatus*-infected lung tissue (Supplementary Fig. 8). Regions of interest (ROI) analysis showed that uptake of $^{18}$F-FDS by the infected lung was 30.7-fold higher than uptake by normal lung (6.98 ± 2.93%ID/g vs. 0.23 ± 0.03%ID/g, $P = 0.0043$). $^{18}$F-FDS PET could also distinguish IPA from lung infection by *S. aureus*, with the latter showing negligible uptake of $^{18}$F-FDS (Fig. 4a and Supplementary Fig. 9). $^{18}$F-FDS signal intensity was 25.7-fold higher in IPA lungs than in lungs infected with *S. aureus* (6.98 ± 2.93%ID/g vs. 0.27 ± 0.06%ID/g, $P = 0.0095$) (Fig. 4c). By contrast, reference imaging with $^{18}$F-FDG could not distinguish IPA from *S. aureus* lung infection (Supplementary Fig. 10), indicating that $^{18}$F-FDS could identify the etiological agent of pulmonary infection. Although the uptake of $^{18}$F-FDS and $^{18}$F-FDG by *A. fumigatus*-infected lungs was comparable (6.98 ± 2.93%ID/g vs. 8.38 ± 1.07%ID/g, $P = 0.3524$, Supplementary Fig. 11), the tracer uptake ratio by infected lung and background tissue (muscle) was significantly higher following $^{18}$F-FDS than $^{18}$F-FDG injection (4.62 ± 1.15 and 0.80 ± 0.07, $P = 0.0095$, Supplementary Fig. 12). The specificity of $^{18}$F-FDS PET for IPA was further verified by assessing its uptake by lung metastases of malignant melanoma (B16F10 cells) and by lipo-polysaccharide (LPS)-induced lung inflammation in mouse models. Pathologically confirmed cancerous lesions and areas of sterile inflammation in the lungs that showed significant $^{18}$F-FDG uptake failed to show significant uptake of $^{18}$F-FDS (Supplementary Figs. 13–15).

Because the central nervous system (CNS) is one of the most common sites of *Aspergillus* infection in immunocompromised hosts, the ability of $^{18}$F-FDS to detect *Aspergillus* was also evaluated in brain infection. Postmortem histopathologic examination of the mouse cerebrum verified the formation of abscesses and the infiltration of *A. fumigatus* hyphae in the right cerebrum (Fig. 5a–c). $^{18}$F-FDS localized to the site of *Aspergillus* brain infection, but no signal was seen in normal brains (Fig. 5d and Supplementary Fig. 16). By contrast, $^{18}$F-FDG signal intensities

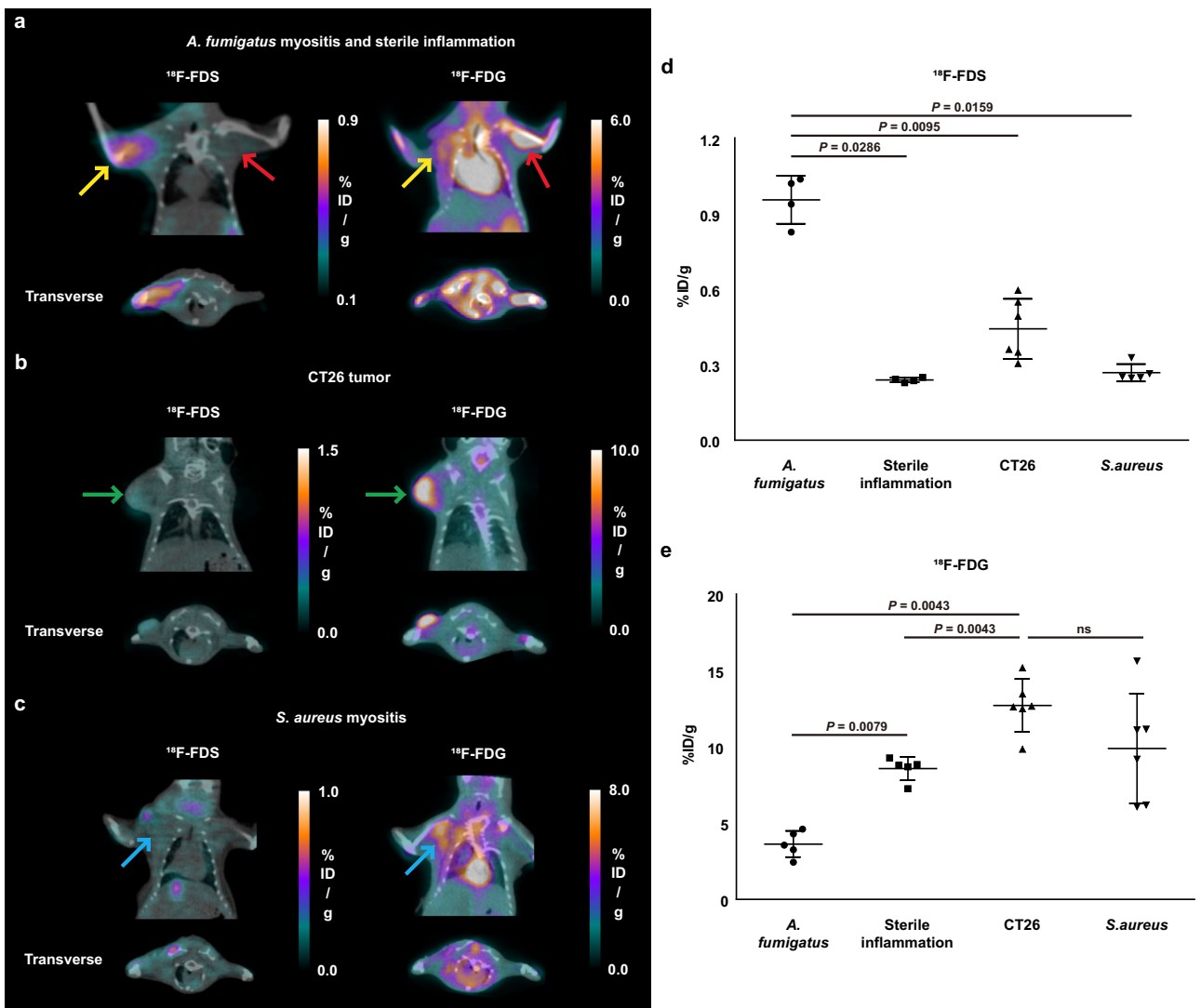

**Fig. 3 MicroPET assessment of $^{18}$F-FDS and $^{18}$F-FDG in BALB/c mice with *A. fumigatus*-infected myositis, sterile inflammation, tumor, and *S. aureus*-infected myositis.** Mice were inoculated in the right shoulder with *A. fumigatus* or *S. aureus* ($1 \times 10^8$) and in the left shoulder with $1 \times 10^7$ heat-killed *A. fumigatus* to induce sterile inflammation. Other mice were inoculated in the right shoulder with $1 \times 10^6$ CT26 cells. **a** MicroPET/CT images of $^{18}$F-FDS and $^{18}$F-FDG in mouse shoulders with *A. fumigatus*-infected myositis (yellow arrow) and sterile inflammation (red arrow) (7.4 MBq, $n \geq 4$ each). **b** MicroPET/CT images of $^{18}$F-FDS and $^{18}$F-FDG in mouse shoulders bearing CT26 tumors (green arrow, $n = 6$ each). **c** MicroPET/CT images of $^{18}$F-FDS and $^{18}$F-FDG in mouse shoulders with *S. aureus*-infected myositis (blue arrow, $n \geq 5$ each). **d**, **e** Quantification of $^{18}$F-FDS (*A. fumigatus*-infected myositis; $n = 4$, sterile inflammation; $n = 4$, CT26 tumor; $n = 6$, and *S. aureus*-infected myositis; n = 5) and $^{18}$F-FDG (*A. fumigatus*-infected myositis; $n = 5$, sterile inflammation; $n = 5$, CT26 tumor; $n = 6$, and *S. aureus*-infected myositis; $n = 6$) radioactivity in shoulder muscles of the above-described mice (ns = not significant). Data are expressed as the mean ± SD. Statistical significance was calculated using two-tailed Mann–Whitney U tests.

did not differ significantly in infected and normal brains (Fig. 5e), with the uptake ratio between infected brain lesions and surrounding tissue being 12.2-fold higher with $^{18}$F-FDS than with $^{18}$F-FDG ($P = 0.0061$; Fig. 5f). Overall, these results suggested that $^{18}$F-FDS was a suitable tracer for localization of *A. fumigatus* brain infection.

**$^{18}$F-FDS PET can monitor antifungal efficacy in *A. fumigatus* infection.** $^{18}$F-FDS PET could be utilized in patients receiving antifungal treatment to evaluate their microbial burden and to monitor the efficacy of treatment[33,34]. $^{18}$F-FDS PET imaging was therefore performed in mice with *A. fumigatus*-infected myositis, before and after treatment with voriconazole (40 mg/kg), a drug of choice for the treatment of IA, or fluconazole (20 mg/kg), a negative control. The $^{18}$F-FDS signal, which corresponded to *Aspergillus* CFU, disappeared after treatment with voriconazole,

but not after administration of fluconazole (Fig. 6). The increases in CFU and $^{18}$F-FDS signal in mice receiving fluconazole corresponded to disease progression. Moreover, these mice appeared sick, with body weight loss, ruffled fur, a hunched posture, reduced movement, and earlier death, with two mice dying 8 days and five dying 13 days after treatment, compared with the voriconazole-treated group (Fig. 6d and Supplementary Fig. 17). Early determination of treatment failure would be helpful in the management of seriously infected patients.

**Discussion**
$^{18}$F-FDS showed selective concentration and prolonged retention in regions of *A. fumigatus* infection with low background activity, suggesting that this radiotracer could be used as a PET molecular imaging agent to obtain high quality images in the diagnosis of *A. fumigatus* infection. $^{18}$F-FDS uptake differentiated *A. fumigatus*

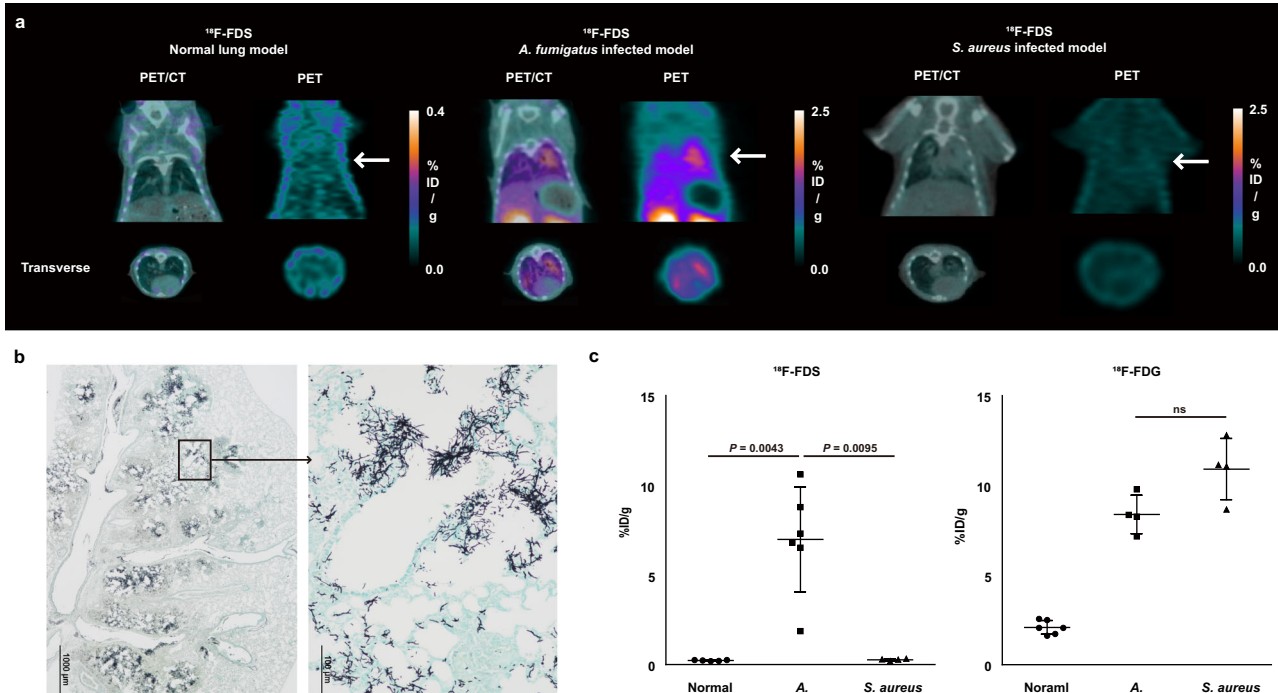

**Fig. 4 MicroPET/CT assessment of $^{18}$F-FDS and $^{18}$F-FDG in BALB/c mice with *A. fumigatus* and *S. aureus* lung infection 2 h post-injection.**
**a** Representative microPET and merged images of $^{18}$F-FDS in normal ($n = 5$), *A. fumigatus* ($n = 6$), and *S. aureus* ($n = 4$) infected lungs (white arrow). **b** *A. fumigatus*-infected lung lobe section stained with methenamine silver 48 h after infection showing the diffuse infiltration of *A. fumigatus* in the airways and lung parenchyma (left, original magnification × 20, n = 4). Magnification of the infiltrated hyphae of *A. fumigatus* (right, original magnification ×200, n = 4). **c** Quantification of the in vivo $^{18}$F-FDS and $^{18}$F-FDG PET images (ns = not significant). Data are expressed as the mean ± SD. Statistical significance was calculated using two-tailed Mann–Whitney U tests.

infection from other infectious and non-infectious causes. These imaging findings correlated well with the clinical course of IA, thus enabling the effects of treatment to be monitored.

$^{18}$F-FDS PET was able to visualize *A. fumigatus* infections of the lungs, brain, and muscles in mouse models of aspergillosis. The in vivo target-to-background ratios were 31.20 ± 12.82 in IPA (infected/normal lungs), 10.19 ± 2.98 in a model of brain infection (infected/surrounding normal brain), and 8.90 ± 1.81 in a model of myositis (infected/normal muscle). This study also showed that $^{18}$F-FDS uptake was specific for *A. fumigatus* infection, differentiating this condition from sterile inflammation, tumors, and Gram-positive bacterial (*S. aureus*) infection in mice. In conventional imaging, such as CT, MRI, and $^{18}$F-FDG PET, pulmonary or CNS aspergillosis is often mistaken for tumors[35–38]. These diagnoses require additional confirmation by histopathology or culture, but these methods are often too invasive and take several days to get results. In patients with risk factors for IPA, pulmonary infiltration is induced by various causes other than aspergillosis. For example, one study in non-HIV immunocompromised patients found that the etiology of pulmonary infiltrates was infectious in 77% and non-infectious in 23% of patients, with the most frequent pathogens being *A. fumigatus* (23%), *S. aureus* (14%), and *Pseudomonas aeruginosa* (10%)[39]. Therefore, distinguishing *A. fumigatus* infection from non-infectious or other infectious causes is critical. The present study verified that $^{18}$F-FDS uptake could differentiate *A. fumigatus* infection from other infectious and non-infectious causes. By contrast, $^{18}$F-FDG uptake could not differentiate *Aspergillus* infection from sterile inflammation, CT26 tumor, and *S. aureus* infection. Especially in the brain aspergillosis model, the infected area was highly localized and visualized by $^{18}$F-FDS, not by $^{18}$F-FDG. One of

the major advantages of $^{18}$F-FDS as a tracer is that it can be synthesized rapidly from $^{18}$F-FDG through simple reduction, which can be easily performed either on-site or at a nearby cyclotron. Therefore, $^{18}$F-FDS PET imaging has potential application in the early diagnosis of IA in clinical settings. Even if $^{18}$F-FDS cannot be prepared as quickly as needed, its high accuracy suggests that $^{18}$F-FDS would be a promising tracer for more delayed diagnosis. Moreover, empirical antifungal therapy, which is usually administered to patients with suspected IA, could be adjusted to definitive therapy based on the results of $^{18}$F-FDS PET scanning.

$^{18}$F-FDS PET scanning after voriconazole treatment of mice with *A. fumigatus*-infected myositis showed that uptake diminished over time. These imaging findings correlated well with the clinical courses of these mice, in that they survived until the end of treatment (16 days post-infection), with improved appearance and increased physical activity. Evaluation of the efficacy of antifungal agents may be difficult in immunocompromised patients with IA. Because the clinical signs and symptoms of IA can be non-specific, use of these clinical parameters to characterize treatment response may lead to erroneous interpretations. Clinical manifestations suggesting ongoing inflammatory reactions can indicate treatment failure in patients receiving antifungal therapy, or they may be related to other factors, including concomitant infection, underlying medical conditions, and immune reconstitution inflammatory syndrome[40]. Conventional radiology, such as CT and MRI, cannot monitor therapeutic responses because they reveal structural abnormalities and inflammation, which do not subside rapidly. Thus, the ability of $^{18}$F-FDS to monitor the course of treatment suggests that this method will play a certain role in clinical decision making in patients with IA.

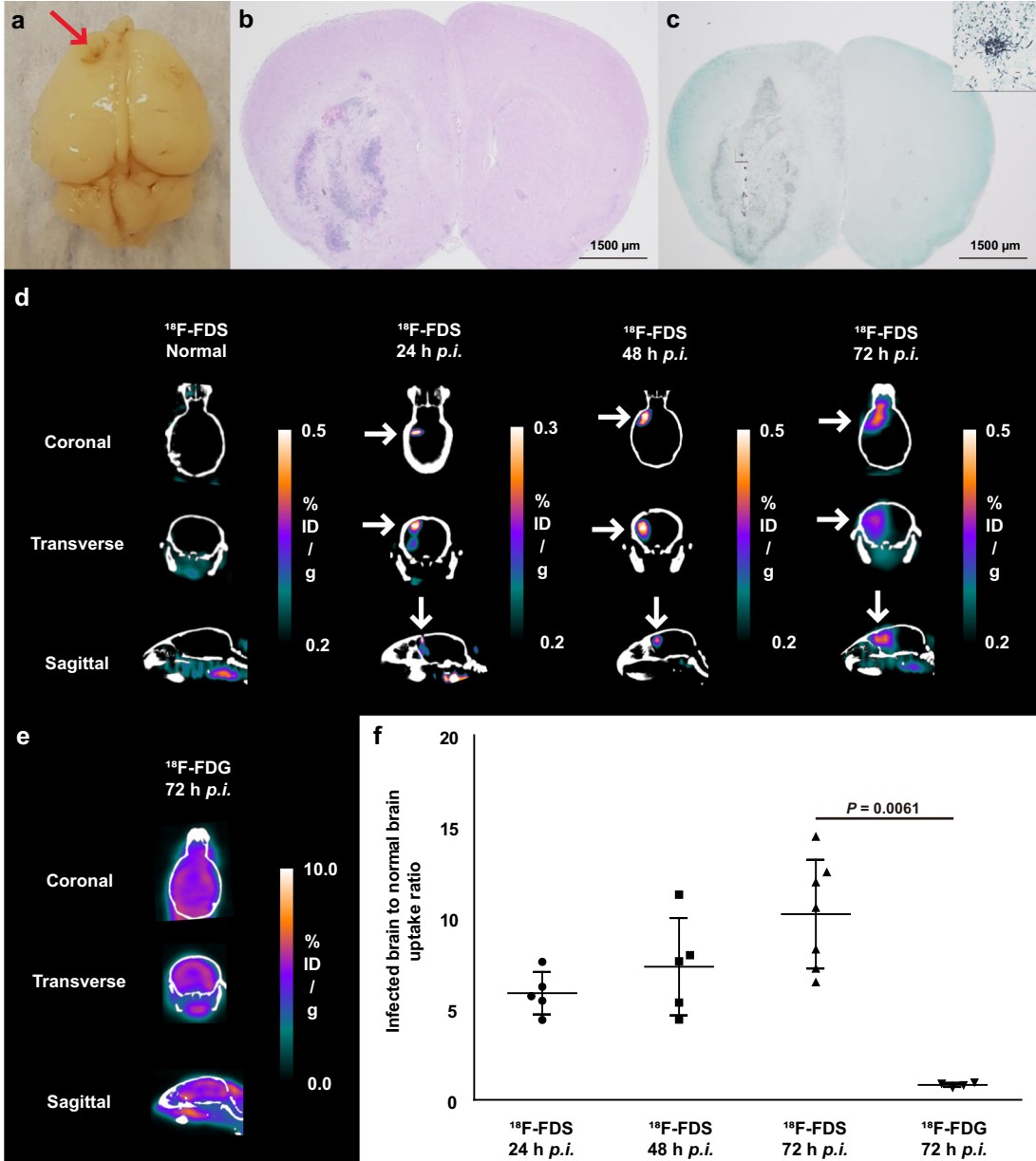

**Fig. 5 MicroPET/CT assessment of $^{18}$F-FDS in female C57BL/6 mice with *A. fumigatus* brain infection. a** Gross view of a mouse brain with stereotactically-injected *A. fumigatus* 3 mm caudal and 3 mm lateral to the right of the bregma and at a 3.5 mm depth from the brain surface 3 days after infection. **b** Hematoxylin and eosin (H&E) staining showing abscess formation in the right cerebrum 3 days after infection (original magnification × 10, *n* = 6). **c** *A. fumigatus*-infected brain section stained with methenamine silver stain (original magnification ×10, *n* = 6). The right upper box shows magnified infiltrated *A. fumigatus* hyphae. **d** Representative microPET/CT images of $^{18}$F-FDS in normal and infected brains at 24 h (*n* = 5), 48 h (*n* = 5), and 72 h (*n* = 7; white arrows) post-infection (*p.i.*). **e** $^{18}$F-FDG images in mice with brain infection at 72 h *p.i.* (*n* = 4). **f** Uptake ratios of $^{18}$F-FDS and $^{18}$F-FDG in infected areas relative to surrounding brain tissue ($^{18}$F-FDS at 24, 48, and 72 h; *n* = 5, 5, and 7, each, $^{18}$F-FDG at 72 h; *n* = 4.) Data are expressed as the mean ± SD. Statistical significance was calculated using two-tailed Mann–Whitney U tests.

$^{18}$F-FDS PET imaging in mice with IPA revealed high background uptake, which had not been previously observed in other studies using animals infected with *E. coli* or *Klebsiella pneumoniae*[26,27]. In the most recent study on $^{18}$F-FDS uptake in *Aspergillus* spp., dynamic PET/CT imaging showed that the radioactivity of $^{18}$F-FDS in the blood appeared to be maintained longer in *A. fumigatus* infected mice than in healthy control mice, which might be due to hemodynamic change, such as systemic vasodilation caused by serious infection[41]. A blood pool effect of $^{18}$F-FDS PET was observed in human patients, suggesting that our findings may also be associated

with a blood pool effect[34]. However, if the increased background uptake of $^{18}$F-FDS was solely related to enhanced blood pool effect, which is the manifestation of sepsis, similar uptake should have been observed in mice with *S. aureus* lung infection that were as sick as IPA mice. Even if pathogen-specific uptake and an increased blood pool effect coexist in IPA mice, the potential of $^{18}$F-FDS PET imaging should not be overlooked, given that in diverse IA models the target-to-background ratio is high in brain and muscle, as well as in lung (4.62 ± 1.15), which is in contrast to $^{18}$F-FDG PET imaging (0.80 ± 0.07, Supplementary Fig. 12).

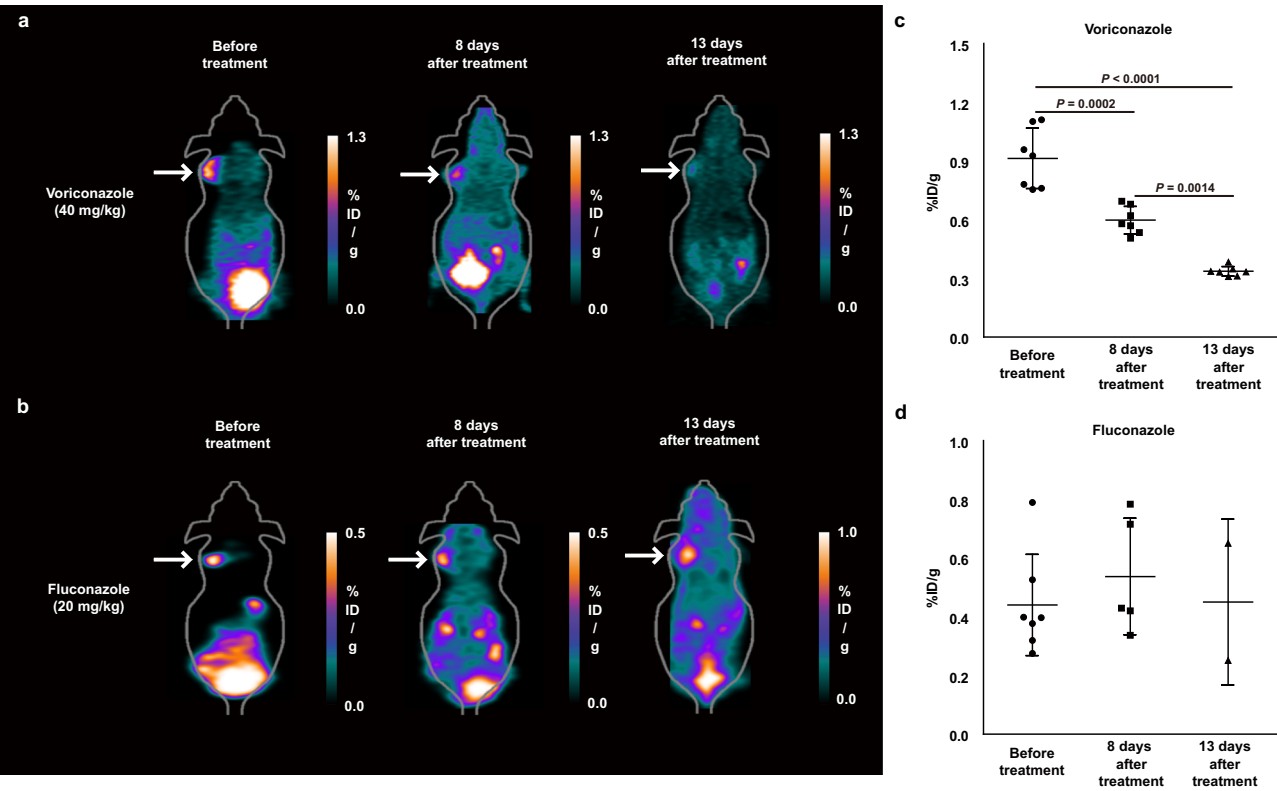

**Fig. 6 ¹⁸F-FDS PET monitoring of the antifungal activity of voriconazole in mice with *A. fumigatus*-infected myositis.** Mice were intraperitoneally injected with voriconazole (40 mg/kg) or, as a control, fluconazole (20 mg/kg) every 24 h from day 3 to day 16 post-infection. Both groups received additional intraperitoneal injections of cyclophosphamide (200 mg/kg) on days 6, 9, 12, and 15 after infection. Representative microPET images of mice treated with voriconazole (**a**) and fluconazole (**b**) before (n = 7 each) and after 8 and 13 days of treatment (white arrow). Quantitative analysis of ¹⁸F-FDS in mice treated with voriconazole ((**c**), n = 7) and fluconazole ((**d**), Before treatment; n = 7, 8 days after treatment; n = 5, 13 days after treatment; n = 2). Data are expressed as the mean ± SD. Statistical significance was calculated using two-tailed Mann–Whitney U tests.

An in vitro uptake assay demonstrated that ¹⁸F-FDS uptake in *A. fumigatus* was as high as in *E. coli* when it was normalized to $1 \times 10^7$ conidia or CFU. However, when in vitro ¹⁸F-FDS uptake of microbes was normalized to protein, significant reduction of uptake was observed in fungi in comparison with *E. coli* (Supplementary Fig. 1). As fungi and bacteria have very different growth patterns, dynamics, and metabolism, determination of a universal indicator to normalize the biomass for both is difficult[41]. In broth media, not all conidia of *A. fumigatus* are activated and germinated to form hyphae, and some remain as dormant conidia, even after relatively long incubation periods[42]. As hyphae take up more sorbitol than conidia[41], if dormant conidia remain in the culture media, the uptake value might be underestimated when normalized with protein. In a further experiment with near infrared fluorescence microscopy, we were able to directly observe accumulation of fluorescent sorbitol (sorbital-ZW800-1) in germinated *A. fumigatus* in comparison with conidia (Supplementary Fig. 2). In most in vivo infected tissue, *A. fumigatus* is observed as well-developed hyphae, as germination is an essential phase in invasive Aspergillus infection[4], and we also observed well-developed hyphae in our histological findings (Figs. 4 and 5 of the manuscript and Supplementary Fig. 5). We subsequently observed significantly higher ¹⁸F-FDS uptake in the infected tissue than in the background tissue on PET imaging. Therefore, we speculate that because of differences in the growth of *A. fumigatus* between biotic and abiotic environments, in vitro uptake might be underestimated when normalized according to the amount of protein.

This study had several limitations. Because ¹⁸F-FDS also accumulates in Enterobacterales[26], ¹⁸F-FDS PET imaging could not differentiate *A. fumigatus* infection from Enterobacterales infection. Clinically, however, IA is mainly diagnosed in immunocompromised hosts and generally manifests as lung, sinus, and CNS infections, whereas Enterobacterales mainly cause intra-abdominal infection regardless of host immune status. The combination of ¹⁸F-FDS PET and appropriate clinical information may have a role in the differential diagnosis and localization of IA. Testing also showed that ¹⁸F-FDS was taken up in vitro by cells of other fungi, including *R. arrhizus* and *C. albicans*. ¹⁸F-FDS PET imaging of these fungi has not been evaluated in vivo to date. In contrast to mold infection, candida is an infrequent cause of pulmonary, brain, and sinus infections, and is relatively easy to diagnose microbiologically. Infection with *Rhizopus*, one of the most frequent causes of invasive mucormycosis, is usually fatal and may be difficult to distinguish from IA. Mucormycosis is increasingly reported related to COVID-19. Although most of these patients had rhino-orbital cerebral disease, some also had pulmonary disease[43]. Additional studies are needed to evaluate the usefulness of ¹⁸F-FDS PET imaging in the diagnosis of *Rhizopus* infection.

Our study shows that ¹⁸F-FDS PET can monitor *A. fumigatus* infection, a leading cause of death in immunocompromised patients. ¹⁸F-FDS has favorable aspects for imaging IA, such as high selectivity for infected tissue and rapid clearance from other organs, leading to superior image quality. In addition, ¹⁸F-FDS is expected to be cost effective, owing to the inexpensive and easy radiosynthesis process. Furthermore, the radioactive decay product of ¹⁸F-FDS, sorbitol, has been approved as a food worldwide, including by the U.S. Food and Drug Administration and the European Food Safety Authority. Thus, ¹⁸F-FDS may be an ideal

candidate for molecular imaging of IA in clinical settings, providing a rapid, noninvasive, and accurate diagnostic tool to identify and localize *Aspergillus* infections, guide the selection of mold-active antifungals, and improve patient outcomes.

In summary, the present study showed that [18]F-FDS has a potential as a tool for non-invasive detection of *A. fumigatus* infections in vivo. The selective concentration and prolonged retention of [18]F-FDS in regions of *A. fumigatus* infection with low background activity suggest that this radiotracer could be used as a PET molecular imaging agent to obtain high quality images in the diagnosis of *A. fumigatus* infection. The excellent and unique characteristics of [18]F-FDS warrant its further investigation for clinical translation.

## Methods

**Synthesis of [19]F-FDS**. [19]F-FDS was prepared by a modification of a previously described method, involving the reduction of [19]F-FDG by sodium borohydride in water[32]. Sodium borohydride (10 mg, 0.265 mmol) was added to a solution of [19]F-FDG (10 mg, 0.055 mmol) in water and the reaction mixture was stirred at 45 °C for 15 min. The mixture was adjusted to pH 6.5–7.5 and passed through a Sep-Pak Alumina N cartridge. The filtrate was evaporated and [19]F-FDS was obtained at a yield of 8.9 mg (87.9%). Parameters included [1]H NMR (300 MHz, water-$d_2$) δ (ppm): 3.47–3.57 (m, 2H), 3.62–3.78 (m, 4H), 3.82–3.96 (m, 1H), 4.37–4.55 (m, 1H), high-resolution mass spectrometry (HRMS, Fast atom bombardment) $m/z$ calculated for $C_6H_{13}FO_5$ [M]$^+$ = 184.0746, actual = 184.0747.

**Preparation of radiotracer**. To prepare [18]F-FDS, [18]F-FDG was synthesized and provided by the Innovation Center for Molecular Probe Development of Chonnam National University Hwasun Hospital. [18]F-FDG was reduced with NaBH$_4$ at 45 °C for 15 min. Its pH was adjusted to pH 6.5–7.5 with acetic acid, and the solution was filtered directly through a Sep-Pak Alumina N cartridge with a sterile Millipore filter (0.22 μm, 4 mm) into a sterile product vial (10 mL size). Finally, instant thin-layer chromatography silica-gel strips (ITLC-SG) of [18]F-FDS, [18]F-FDG, and a spiked sample were developed with a 1:4 solution of water:acetonitrile, and the results were compared for identification of [18]F-FDS (R$_f$ = 0.45 for [18]F-FDS, R$_f$ = 0.67 for [18]F-FDG).

**In vitro uptake assays**. *Escherichia coli* (ATCC 25922), *S. aureus* (ATCC 29213), *A. fumigatus* (isolated from a sputum sample from a patient admitted to the oncology ward of Chonnam National University Hwasun Hospital), *R. arrhizus* (clinical isolate), and *C. albicans* (SC5314) were used for in vitro uptake assays. *E. coli* and *S. aureus* were aerobically grown in Lysogeny Broth (LB) overnight at 37 °C; 1 mL of each sample was transferred to fresh LB and incubated for another 2 h at 37 °C. Colonies were counted and $1 \times 10^7$ CFU of *E. coli* and *S. aureus* were prepared for probe uptake assay. *C. albicans* was cultured on yeast extract peptone dextrose (YPD) plates at 37 °C for 48 h, harvested using sterile phosphate-buffered saline (PBS), and suspended to a final concentration of $1 \times 10^7$ CFU. *A. fumigatus* and *R. arrhizus* were cultured on potato dextrose agar plates for 5–7 days at 30 °C, and colonies were harvested using saline containing 0.1% (v/v) Tween 20. The suspensions were vortexed to release the conidia and filtered through a 40 μm cell strainer to remove clumps and hyphae. Colonies were counted using a hemocytometer, and $1 \times 10^7$ *A. fumigatus* conidia were prepared for the probe uptake assay. The control consisted of *A. fumigatus* heat-killed in an autoclave at 121 °C for 15 min[44]. Probe uptake assays were performed by incubating bacterial or fungal cultures with [18]F-FDS (0.74 MBq/mL) at 37 °C with rapid agitation (200 rpm). Bacteria or fungi were filtered using micro centrifuge tube filters and washed three times with PBS. The radioactivity of each supernatant and pellet was measured using a gamma counter.

To present the Bq/μg of protein in in vitro uptake studies, $1 \times 10^6$ CFU of cell pellets suspended in 0.5 mL of Y-PER™ yeast protein extraction reagent (Thermo-Fisher Scientific, 78991) or B-PER™ bacterial protein extraction reagent (Thermo-Fisher Scientific, 78243) were mixed with Protease Inhibitor Cocktail Solution (GenDEPOT, P3100-010) and transferred to 2 mL reinforced tubes (Precellys, P000943-LYSK0-A.O) containing 200 μL dry volume of 212–300 μm glass beads (Sigma-Aldrich, G9143). The mixture was incubated on ice for 30 min, and the cells were disrupted by five 65 s blasts of vigorous vortexing, interspersed with 2 min incubations of the tube on ice. Following the final vortexing, the beads were allowed to settle and each supernatant was decanted into a fresh tube. Protein concentrations were quantified using BCA™ Protein Assay kits (Thermo-Fisher Scientific, 23225).

**Confocal microscopy**. *A. fumigatus* was cultured on potato dextrose agar plates for 5–7 days at 30 °C, and colonies were harvested using PBS containing 0.1% (v/v) Tween 20. The suspensions were vortexed to release the conidia and filtered through a 40 μm cell strainer to remove clumps and hyphae. Conidia were counted using a hemocytometer, and $1 \times 10^6$ *A. fumigatus* conidia and 50 μM of sorbitol-

ZW800-1 were co-cultured for up to 48 h in 1 mL of YPD broth in 24-well plates at 37 °C without agitation. Sorbitol-ZW800-1 was prepared as described previously[45]. Briefly, to prepare sorbitol-ZW800-1, ZW800-1 NHS ester (1 mmol, 1 mg) was conjugated with amino-sorbitol (1.5 mmol, 0.27 mg) using N,N-diisopropylethyl amine (5 mmol, 0.65 mg) in DMSO (2 mL) at room temperature for 1 h. The crude mixture was purified by a preparative HPLC system. Co-cultured *A. fumigatus* were observed at 20 and 48 h using laser scanning confocal microscopy (LSM 800, ZEISS, Germany). Before microscopic examination, cultured *A. fumigatus* were washed twice, re-suspended in PBS, and transferred to a confocal dish. Fluorescence of ZW800-1 (red) was obtained using laser filter excitation at 561 nm and emission at 650–750 nm. Image acquisition and analysis were performed using ZEN 2.6, blue edition (ZEISS, Germany).

**Animal models and biodistribution studies**. Animal care, animal experiments, and euthanasia were performed in accordance with protocols approved by the Chonnam National University Animal Research Committee and the Guide for the Care and Use of Laboratory Animals.

BALB/c mice (female, 8 weeks old) were used for muscle and lung infection; C57BL/6 mice (female, 6 weeks old) were used for brain infection and B16F10 (mouse melanoma cell line) lung metastases; and C57BL/6 (male, 8 weeks) mice were used for LPS-induced acute lung injury. *A. fumigatus*-infected myositis models have been described[46]. Briefly, BALB/c mice were immunosuppressed by intraperitoneal injections of cyclophosphamide (200 mg/kg) 4 and 1 days prior to inoculation and a subcutaneous injection of cortisone acetate (125 mg/kg) 1 day prior to inoculation, followed by inoculation of the right shoulder muscle with $1 \times 10^8$ *A. fumigatus* conidia. [18]F-FDS scans were performed 72 h after infection.

To evaluate the correlation between [18]F-FDS uptake and number of cells, mice were inoculated in their right shoulders with $1 \times 10^1$ to $1 \times 10^8$ *A. fumigatus* conidia. Following [18]F-FDS imaging, the mice were sacrificed and their infected right shoulder muscles were weighed and homogenized in 5 mL of sterile saline. Homogenates were serially diluted and spread on potato dextrose agar plates, which were incubated at 30 °C for 48 h. *A. fumigatus* CFU were counted, with the data expressed as log CFU/g of tissue. The minimum number of *A. fumigatus* conidia that could be visually identified on [18]F-FDS PET was determined by ROC curve analysis. [18]F-FDS uptake by infected and normal muscle was compared in each subgroup, divided by the cutoff number of *A. fumigatus* conidia.

The *S. aureus*-infected myositis model was generated by injecting $1 \times 10^8$ CFU *S. aureus* into the right shoulder muscles of sedated 8-week-old female BALB/c mice[29]. Sterile inflammatory myositis was induced by injection into the right shoulder muscles of heat-killed conidia[44]. To generate tumor xenografts, CT26 (mouse colon carcinoma) or B16F10 cells were cultured in DMEM supplemented with 10% fetal bovine serum (FBS) at 37 °C in a 5% CO$_2$ incubator. CT26 cells ($1 \times 10^6$) were injected into the right shoulder muscles of female, 6-week-old BALB/c mice. The lung metastases models were generated by i.v. injection of B16F10 cells ($2 \times 10^5$)[47]. After 10 days, the mice were subjected to microPET imaging.

To generate *A. fumigatus*-infected lungs, BALB/c mice were immunosuppressed by intraperitoneal injection of cyclophosphamide (150 mg/kg) 4 days prior to inoculation and subcutaneous injection of cortisone acetate (250 mg/kg) 1 day prior to inoculation, followed by intranasal inoculation with $1 \times 10^8$ *A. fumigatus* conidia[48]. *S. aureus*-associated pneumonia was induced in sedated 8-week-old female BALB/c mice by intranasal instillation of 40 μl of PBS containing $2 \times 10^8$ CFU *S. aureus*[49]. [18]F-FDS scans were performed 48 h after infection unless otherwise indicated. A mouse model of acute sterile lung inflammation[50] was induced by intraperitoneal injection of LPS (20 mg/kg, *E. coli* 055:B5, Sigma-Aldrich; St. Louis, MO) into C57BL/6 mice, with [18]F-FDS imaging performed 4 h after LPS injection.

For brain infection, C57BL/6 mice were immunosuppressed with 200 mg/kg of cyclophosphamide 4 days prior to infection. On the day of brain infection, the mice were anesthetized with a 4:1 mixture of alfaxalone:xylazine and fixed in a stereotactic frame. The midline scalp was dissected and a small hole (3 mm caudal and 3 mm lateral to the right of the bregma) was drilled into the skull of each mouse. Using a 10 μL Hamilton syringe, a suspension of $1 \times 10^5$ *A. fumigatus* conidia in 3 μl saline was injected through the hole 3.5 mm deep into the brain of each mouse. Immediately after retraction of the needle, the hole in the skull was sealed with bone wax and the scalp was sutured[51]. [18]F-FDS scans were performed 24 h, 48 h, and 72 h after infection.

The biodistribution of [18]F-FDS in different organs was assessed in mice with *A. fumigatus*-infected myositis 2 h after intravenous injection of 7.4 MBq of radiotracer. After sacrifice, the blood, heart, lungs, liver, spleen, stomach, intestines, kidneys, pancreas, normal muscles, bones, brain, skin, and infected muscles were extracted from the mice and weighed. The radioactivity of each organ was determined using a gamma counter, with these measurements of radioactivity normalized relative to the weight of each organ sample and the amount of radioactivity injected to obtain the % ID/g (ID: injected dose).

**Antifungal efficacy**. To measure the efficacy of antifungal treatment, mice infected with *A. fumigatus* on the right shoulder were intraperitoneally injected with 40 mg/kg of voriconazole (Pfizer) every 24 h from day 3 to day 16 post-infection. Control mice were similarly injected with 20 mg/kg fluconazole (Pfizer) every 24 h from day 3 to day 16 post-infection. All mice also received intraperitoneal

injections of cyclophosphamide (200 mg/kg) on days 6, 9, 12, and 15 post-infection. MicroPET scans were performed on post-infection days 3 (before treatment), 11 (after 8 days of treatment), and 16 (after 13 days of treatment). Following intravenous injection of 7.4 MBq [18]F-FDS, microPET images were acquired for 10 min at 2 h.

**Small animal PET/CT examination.** PET images were obtained 48 h after inoculation of *A. fumigatus* or *S. aureus* into mouse lungs; 24, 48, and 72 h after inoculation of *A. fumigatus* into brain; and 72 h after inoculation of *A. fumigatus* or *S. aureus* into muscle. All images were acquired using a high-resolution small animal PET-SPECT-CT scanner (Inveon, Siemens Medical Solutions). One or two hours after intravenous injection of 7.4 MBq [18]F-FDS or [18]F-FDG, micro-PET/CT images were acquired for 10 min. To compare [18]F-FDS and [18]F-FDG images of CT26 tumors, mice with these tumors were injected with [18]F-FDS 1 day after [18]F-FDG microPET imaging. ROI were drawn manually over the infected organs, normal organs, tumor, and areas of inflammation on decay-corrected coronal images, using CT images as a guide. Acquired images were reconstructed with a three-dimensional ordered subset expectation maximization (OSEM3D) algorithm. MicroPET image analysis was performed with PMOD software (PMOD Technologies Ltd, Zürich, Switzerland). Radiotracer uptake was expressed as %ID/g.

**High-resolution autoradiography.** Mice with *A. fumigatus*-infected lungs were injected intravenously with 111 MBq of [18]F-FDS and sacrificed. Their lungs were extracted, frozen in isopentane, and cut into 20 μm sections using a cryostat. The sections were thaw-mounted onto glass microscope slides, which were exposed to the phosphor screen overnight and scanned in a Typhoon FLA 9500 scanner (GE Healthcare, USA). Photographs of the sections were recorded with a digital camera.

**Statistical analysis.** All data are expressed as mean ± standard deviation (SD). Differences in continuous variables were evaluated using Mann–Whitney U tests. The optimal cutoff for *A. fumigatus* conidia was obtained using ROC curve analyses. All analyses were performed with GraphPad Prism 8.0 software (GraphPad Software Inc., San Diego, California, USA), with $P$ values $< 0.05$ considered statistically significant.

**Reporting summary.** Further information on research design is available in the Nature Research Reporting Summary linked to this article.

## Data availability

Data supporting the findings described in this study are available within the article and its Supplementary Information. Additional data related to this study are available from the corresponding authors upon request. Source data are provided with this paper.

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

## Acknowledgements

We thank Hwa Youn Jang, Myeongjun Kim, and Jeong hwan Kwon (Innovation Center for Molecular Probe Development) for their excellent research assistance. This study was supported by the Commercialization Promotion Agency for R&D Outcomes (COMA) grant (2021C300) and National Research Foundation grants (2015M3C1A3056410, 2021M3C1C3097637) funded by the Ministry of Science and ICT (MSIT) and the Technology Innovation Program (20016670) funded By the Ministry of Trade, Industry & Energy (MOTIE, Korea). D.Y.K. was supported by a National Research Foundation of Korea (NRF) grant funded by MSIT (NRF-2020R1C1C1012379 and NRF-2021M2E7A3085685) and A.P. was supported by the Basic Science Research Program through the NRF of Korea funded by the Ministry of Education (NRF-2020R1I1A1A01070543). S.J.K. was supported by a grant from the Korea Health Technology R&D Project through the Korea Health Industry Development Institute (KHIDI), funded by the Ministry of Health & Welfare, Republic of Korea (HI20C0079040020), and a grant from Chonnam National University Hwasun Hospital Biomedical Institute (HCRI18013-1). U.J.K. was supported by the Bio & Medical Technology Development Program of the NRF funded by the Korean government (NRF-2019M3E5D1A02067959). H.B.L was supported by a grant (2021-4089) from Chonnam National University.

## Author contributions

D.Y.K. and A.P. contributed equally. S.J.K. and J.J.M. directed and coordinated this study. D.Y.K., A.P., S.J.K. and J.J.M. designed and analyzed data. D.Y.K., A.P. designed the experiments. D.Y.K., A.P., S.J., S.H.Y., H.H., S.Y.K. conducted the experiments. S.E.K., D.L., H.K., S.J.O., Y.R.J. involved in animal experiments. D.Y.K., A.P., S.H.Y., and S.J. acquired data. K.H.L, U.J.K, S.Y.K, S.R.K., H.B.L, K.S.M, and S.L. provided critical comments on this project. D.Y.K., A.P., S.J.K. and J.J.M. wrote and reviewed the manuscript. All authors read, commented on and approved this manuscript.

## Competing interests

The authors declare no competing interests.
