## [Peer Review File · Nature Communications]

Reviewers' Comments:

Reviewer #1:

Remarks to the Author:

This study by Kim et al assessed noninvasive detection of *Aspergillus* infections using 2-deoxy-2-¹⁸F-fluorosorbitol (18F-FDS) positron emission tomography (PET). The authors demonstrate that 18F-FDS PET visualized *A. fumigatus* infection in several mouse models distinguishing it from *S. aureus* infections (lung), tumor (brain) and sterile inflammation (muscle). Some suggestions and clarifications to improve the manuscript are listed below:

1. The in vitro uptake data (Figure 1) comparing fungi and bacteria is presented is per 10⁷ cells. However, fungal cells are substantially bigger than bacteria (~100-500 fold in volume) and subsequently may take up more tracer per cell. Therefore, this data needs to be presented per mg of protein.
2. Given that the fungi can replicate over time, the *A. fumigatus* infection burden should be measured after imaging? While the authors state this, these data were not shown. This is especially important for the limit of detection studies (Figure 2b).
3. Clinically, invasive aspergillosis usually presents with multiple hypermetabolic nodules on 18F-FDG PET. However, in Figure 3, there is no 18F-FDG uptake at the site of *Aspergillus* infections. Why is this? Interestingly, 18F-FDG PET uptake is high in pulmonary *Aspergillus* infections (Figure 4c). These findings need to be clearly explained.
4. 18F-FDS PET imaging demonstrated ~9-fold higher uptake in infected muscle versus the contralateral normal muscle. But biodistribution studies (Table 1) show < 2-fold difference. Can the authors explain this?
5. In Figure 4a, the 18F-FDS PET data needs to be shown on the same scale for the panels showing normal, *Aspergillus*-infected and *S. aureus*-infected lungs. Otherwise, these data may be misleading.
6. Since invasive pulmonary aspergillosis (IPA) is one of the primary forms of this disease, the experiments with *S. aureus* are useful and supportive, it would be important to show that this technique shows specificity in distinguishing IPA from a cancerous lesions or sterile inflammation in the lungs. Additionally, it would be important to provide data from dynamic 18F-FDS PET to understand whether the PET signal in IPA is independent of blood pool effect.
7. There is high 18F-FDS PET background in the *Aspergillus*-infected animals (Figure 4a). This is unusual for 18F-FDS PET and the authors need to explain this.
8. While the autoradiography data is useful, the whole lung is lighting up. Can the authors show data where uninfected area within the same lung are included?
9. In Figure S10, what tissue was used as the background?

Other points:

1. Due to changes in nomenclature, it would be better to use "Enterobacterales" instead of "Enterobacteriaceae".
2. It would be important to cite other reports about FDS PET for infection imaging (PMID: 28848037, 27635025) including published clinical studies showing safety and efficacy (PMID: 27020679, 33853931)
3. While 18F-FDS has been evaluated for its ability to detect gliomas (Li et al. Mol. Imaging Biol. 2008), it was not found to be a good imaging agent for this indication. Please modify the introduction to clarify this, as it sounds misleading as written currently.

4. Why is there no corresponding CT for panel b in Figure 3 and in Figure 6? This should be shown.
5. It is unclear (Figure 5) what the 24, 48 and 72 hours time-points mean.
6. Table 1 and Table S1 seem to be identical. If so, please remove one of the tables.

Reviewer #2:

Remarks to the Author:

Innovative study evaluating in vitro and in vivo the new PET tracer fluorodeoxysorbitol for detecting invasive aspergillosis and specifically the differentiating between aspergillosis from other types of infection and cancer.

The study is well designed and clearly written.

Some major comments:

- in the different in vitro and in vivo studies the uptake differences between fungal species, tumor model and S.aureus model are clearly described; the results of W. coli are mostly missing, and this Gram negative bacterial type is of specific interest since it will also take up sorbitol.
- the number of mice used in the different studies is unclear. If I'm correct then sometimes the number is only 3 or 4; not enough to draw significant conclusions.
- the discussion should be rewritten; the first part (line 164-200) is more a repetition of the introduction. The start of the discussion should focus on the important findings of this study.
- In the discussion a part should be added how to implement this tracer in the clinics; especially since the authors also mention that there is only a narrow window for diagnosis. It will be difficult for most nuclear centers to produce F-sorbitol within a short time frame.
- Parts of the introduction can be shortened.
- The mentioned values of FDG uptake (0.892 - 0.372 etc) are very low; are these SUV calculations, ratios, MBq/cc. This should be further elaborated
- The myositis model: is that really an infection? Myositis is an inflammatory disease; the authors injected A.fumigatus in the right shoulder; then this is not myositis but a fungal infection focus in the soft tissue of the right shoulder
- Limitations are missing in the discussion

Reviewer #3:

Remarks to the Author:

This spectacular paper reports on in vivo imaging of invasive aspergillosis using 2-deoxy-2-18F-fluorosorbitol (18F-FDS) PET, which clearly visualized A. fumigatus infection of the lungs, brain, and muscles in mouse models. Even more spectacular, this PET imaging did also show promise in assessing treatment success, opening doors for new imaging technologies for treatment monitoring in these often difficult to treat infections.

Comments:

- In the introduction in the section for COVID associated aspergillosis it would be good to mention the Guidance on classification and treatment published recently in Lancet ID by Koehler et al [https://doi.org/10.1016/S1473-3099\(20\)30847-1](https://doi.org/10.1016/S1473-3099(20)30847-1)
- An important limitation was that the 18F-FDS uptake by A. fumigatus, R. arrhizus, and C. albicans was comparable to its uptake by E. coli, so 18 F-FDS may not be useful for differentiating different pulmonary mold infections (aspergillosis versus Mucormycosis) and may also not

differentiate mold infection from gram negative infection. Please discuss more extensively, also considering the emergence of COVID associated Mucormycosis. See Preprint here with 80 cases including many pulmonary cases https://papers.ssrn.com/sol3/papers.cfm?abstract_id=3844587.
- The discussion starts very broadly. Its fine to do it this way but would consider to add a paragraph first that shortly summarizes the most important findings of your study specifically.

REVIEWER COMMENTS

Reviewer #1 (Remarks to the Author):

This study by Kim et al assessed noninvasive detection of *Aspergillus* infections using 2-deoxy-2-¹⁸F-fluorosorbitol (¹⁸F-FDS) positron emission tomography (PET). The authors demonstrate that ¹⁸F-FDS PET visualized *A. fumigatus* infection in several mouse models distinguishing it from *S. aureus* infections (lung), tumor (brain) and sterile inflammation (muscle). Some suggestions and clarifications to improve the manuscript are listed below:

1. The *in vitro* uptake data (Figure 1) comparing fungi and bacteria is presented is per 10⁷ cells. However, fungal cells are substantially bigger than bacteria (~100-500 fold in volume) and subsequently may take up more tracer per cell. Therefore, this data needs to be presented per mg of protein.

(Answer) We again performed *in vitro* uptake experiments and calculated accumulation as Bq/μg of protein. Because fungal cells are substantially larger than bacterial cells, uptake patterns by fungi and bacteria have been reported in Supplementary Figure 1.

Supplementary Figure 1. Cellular uptake of ^{18}F -FDS by *A. fumigatus*, *R. arrhizus*, *C. albicans*, *S. aureus* (-), and *E. coli* (+) cells 1 h after treatment with ^{18}F -FDS.

- Given that the fungi can replicate over time, the *A. fumigatus* infection burden should be measured after imaging? While the authors state this, these data were not shown. This is especially important for the limit of detection studies (Figure 2b).

(Answer) We calculated the CFU at the site of *A. fumigatus* infection immediately after imaging analyses. This has been mentioned in the legend to Figure 2, and CFU data were added to the “Source data” file.

(Page 30, Lines 543–544) The mice were subsequently sacrificed and the CFU were counted immediately.

- Clinically, invasive aspergillosis usually presents with multiple hypermetabolic nodules on ^{18}F -FDG PET. However, in Figure 3, there is no ^{18}F -FDG uptake at the site of Aspergillus infections. Why is this? Interestingly, ^{18}F -FDG PET uptake is high in pulmonary Aspergillus

infections (Figure 4c). These findings need to be clearly explained.

(Answer) After we injected *A. fumigatus* into the shoulder muscle of immunosuppressed BALB/c mice, we observed the progression of necrosis at the injection site over time (Figure R1, below). This finding was not observed following infection with *S. aureus* or the induction of sterile inflammatory myositis by heat killed *A. fumigatus* conidia. Necrotic lesions may explain the low ^{18}F -FDG uptake by mice with *A. fumigatus* myositis mice. ^{18}F -FDG PET imaging in human fungal infection has shown similar findings and has been explained by necrosis of infected organs^{1,2}.

Figure R1. Necrosis of an *A. fumigatus* injected muscle.

Histopathologic findings may also explain the low ^{18}F -FDG uptake. The presence of *A. fumigatus* hyphae in infected muscle is often accompanied by abundant neutrophil infiltration, as shown in Supplementary Figure 4. In some mice, however, aggregated fungal hyphae in infected muscle were accompanied by little inflammatory cell infiltration (Figure R2, below). The latter finding may have been due to the use of cyclophosphamide prior to infection. ^{18}F -FDG PET imaging in infectious and inflammatory diseases is based primarily on the

increased glucose uptake by activated inflammatory cells. Thus, the low uptake of ^{18}F -FDG in some myositis models may be due to the low level of inflammatory cell infiltration into *A. fumigatus*-infected muscle. In addition, we observed increased vascularization at the periphery of infected foci (magnified view of Figure R2), which might explain the increased uptake of ^{18}F -FDG in peripheral lesions of mice with *A. fumigatus* myositis.

Figure R2. Histopathological examination of a mouse with *A. fumigatus*-infected myositis.

Figure 3a in the previously submitted manuscript showed relatively low ^{18}F -FDG uptake in *A. fumigatus* infected myositis ($1.17 \pm 0.30\% \text{ID/g}$), although it was higher than in normal muscle: ($0.56 \pm 0.22\% \text{ID/g}$). Because the original Figure 3a might cause misunderstanding of the data, the present revised manuscript shows a new Figure 3a, demonstrating accumulation of ^{18}F -FDG in *A. fumigatus* infected muscle (Figure 3a, below). The description in the “Results” section has been changed to:

(Page 6, Lines 109–113) By contrast, reference imaging showed that ^{18}F -FDG accumulated not only in *Aspergillus*-infected shoulders, but also in sterile inflamed, *S. aureus*-infected, and tumor-engrafted shoulders (Fig. 3a-c). ^{18}F -FDG uptake by shoulders with sterile inflammation was higher than that by shoulders with *A. fumigatus* infection ($8.57 \pm 0.77\% \text{ID/g}$ vs. $3.63 \pm 0.86\% \text{ID/g}$, Fig. 3e).

4. ^{18}F -FDS PET imaging demonstrated ~9-fold higher uptake in infected muscle versus the contralateral normal muscle. But biodistribution studies (Table 1) show < 2-fold difference. Can the authors explain this?

(Answer) Thank you for this pointing out. We again evaluated the biodistribution of ^{18}F -FDS in mice with *A. fumigatus*-infected myositis (n = 4). We replaced Table 1 and revised the results and numbers of mice in the manuscript as follows.

(Page 6, Lines 114–117) Assessment of *in vivo* biodistribution 2 h after tracer injection also showed selective concentration of ^{18}F -FDS in shoulders with *A. fumigatus*-infected myositis, with concentrations being 10.7-, 2.7-, 8.3-, 14.7-, and 7.2-fold higher than in blood, lungs, liver, brain, and normal muscle, respectively (Table 1).

(Page 37, Lines 599) **Table 1.** Biodistribution of ^{18}F -FDS (2 h, 7.4 MBq, n = 4) in mice with *A. fumigatus*-infected myositis.

Organs	%ID/g (2 h)
Blood	0.31 ± 0.04
Heart	1.00 ± 0.05
Lung	1.20 ± 0.12
Liver	0.40 ± 0.12
Spleen	2.28 ± 0.25
Stomach	0.73 ± 0.25
Intestine	1.92 ± 0.39
Kidney	0.89 ± 0.20
Pancreas	1.92 ± 0.33
Bone	0.70 ± 0.09
Brain	0.23 ± 0.09
Skin	0.57 ± 0.10
Normal muscle	0.46 ± 0.08
Infected muscle	3.32 ± 0.36

5. In Figure 4a, the ^{18}F -FDS PET data needs to be shown on the same scale for the panels showing normal, *Aspergillus*-infected and *S. aureus*-infected lungs. Otherwise, these data may be misleading.

(Answer) We have changed the scale bars of *A. fumigatus* and *S. aureus* images in Figure 4a to match each other (0–2.5). However, this was impossible for the ^{18}F -FDS image of normal lung because the SUV_{max} value of ^{18}F -FDS was only 0.43.

6. Since invasive pulmonary aspergillosis (IPA) is one of the primary forms of this disease, the experiments with *S. aureus* are useful and supportive, it would be important to show that this technique shows specificity in distinguishing IPA from a cancerous lesions or sterile inflammation in the lungs. Additionally, it would be important to provide data from dynamic ¹⁸F-FDS PET to understand whether the PET signal in IPA is independent of blood pool effect.

(Answer) As suggested, we evaluated the specificity of ¹⁸F-FDS PET for IPA compared with lung metastases of malignant melanoma (B16F10) cells and lipopolysaccharide (LPS)-induced lung inflammation. We added the appropriate sentences to the “Results” and “Methods” sections, as shown below, and to Supplementary Figures 12–14. We also performed dynamic ¹⁸F-FDS PET imaging in mice with *A. fumigatus*-infected myositis, enabling us to clearly visualize the sites of infection. ¹⁸F-FDS signals were observed consistently at the infection site through 120 min, whereas blood pool activity from the heart and liver dissipated after 60 min. This finding verified that the ¹⁸F-FDS PET signal in IPA was independent of the blood pool effect.

(Page 7, Lines 137–141) The specificity of ^{18}F -FDS PET for IPA was further verified by assessing its uptake by lung metastases of malignant melanoma (B16F10 cells) and by lipopolysaccharide (LPS)-induced lung inflammation in mouse models. Pathologically confirmed cancerous lesions and areas of sterile inflammation in the lungs that showed significant ^{18}F -FDG uptake failed to show significant uptake of ^{18}F -FDS (Supplementary Fig. 12-14).

Supplementary Figure 12. MicroPET/CT fusion images of ^{18}F -FDS and ^{18}F -FDG in mice with B16F10 lung metastases (7.4 MBq).

Supplementary Figure 13. MicroPET, CT, and PET/CT fusion images of ^{18}F -FDS and ^{18}F -FDG in

mice with LPS-induced lung inflammation (7.4 MBq).

Supplementary Figure 14. Pathological examination of a mouse with LPS-induced acute lung injury (4 hours after intraperitoneal LPS injection). (Left) Hematoxylin and eosin (H&E) staining showing lung alveoli engorged with red blood cells (arrows) (original magnification $\times 100$). (Right) Magnified view of the box in the left panel, showing the accumulation of neutrophils along the bronchioles.

(Page 15, Lines 301–304) BALB/c mice (female, 8 weeks old) were used for muscle and lung infection; C57BL/6 mice (female, 6 weeks old) were used for brain infection and B16F10 (mouse melanoma cell line) lung metastases; and C57BL/6 (male, 8 weeks) mice were used for LPS-induced acute lung injury. *A. fumigatus*-infected myositis models have been described⁴⁴.

(Pages 4-5, Lines 80–84) Dynamic ¹⁸F-FDS PET imaging (over 120 min) in mice with *A. fumigatus*-infected myositis was also performed (Supplementary Movie). ¹⁸F-FDS PET signals could be seen consistently in the infected site even at 120 min, whereas blood pool activities in the heart and liver dissipated after 60 min. This finding indicated that the ¹⁸F-FDS PET signal in *A. fumigatus* infection is independent of blood pool effects.

(Page 15-16, Lines 319–324) To generate tumor xenografts, CT26 (mouse colon carcinoma) or B16F10 cells were cultured in DMEM supplemented with 10% fetal bovine serum (FBS) at 37 °C in a 5% CO₂ incubator. CT26 cells (1×10^6) were injected into the right shoulder

muscles of female, 6-week-old BALB/c mice. The lung metastases models were generated by *i.v.* injection of B16F10 cells (2×10^5)⁴⁵. After 10 days, the mice were subjected to microPET imaging.

(Page 16, Lines 330–333) A mouse model of acute sterile lung inflammation⁴⁸ was induced by intraperitoneal injection of LPS (20 mg/kg, *E. coli* 055:B5, Sigma-Aldrich; St. Louis, MO) into C57BL/6 mice, with ¹⁸F-FDS imaging performed 4 hours after LPS injection.

7. There is high ¹⁸F-FDS PET background in the *Aspergillus*-infected animals (Figure 4a). This is unusual for ¹⁸F-FDS PET and the authors need to explain this.

(Answer) Among mouse models with infection (*A. fumigatus* and *S. aureus*), inflammation, and tumor, only IPA models showed significantly high background activity of ¹⁸F-FDS. We cannot clearly explain this finding at this time, but it may be due to dissemination of *A. fumigatus* to other organs. The respiratory system is the most common portal of entry for *A. fumigatus*, but systemic aspergillosis is not uncommon in immunocompromised hosts^{3, 4}. Mice with lung infections of *A. fumigatus* were sicker than those with muscle or brain infection, sufficient to suggest systemic infection.

¹⁸F-FDS PET showed a blood pool effect in human patients, suggesting our findings may also be related to blood pool effects⁵. However, if the increased background uptake of ¹⁸F-FDS was solely related to enhanced blood pooling, a manifestation of sepsis, similar uptake would have been observed following *S. aureus* lung infection, as these mice were as sick as those with *A. fumigatus* lung infection mice. Although we could not analyze other organs of mice with *A. fumigatus*-associated lung infection histopathologically or microbiologically, disseminated *A. fumigatus* infection may have been responsible for the increased background uptake of ¹⁸F-FDS. Further studies are needed to confirm our speculation.

We added this explanation to the “Discussion” section.

(Page 10, Lines 209–215) ^{18}F -FDS PET imaging in mice with IPA revealed high background uptake, which had not been previously observed in other studies using animals infected with *E. coli* or *Klebsiella pneumoniae*^{27, 28}. A blood pool effect of ^{18}F -FDS PET was observed in human patients, suggesting that our findings may also be associated with a blood pool effect³⁵. However, if the increased background uptake of ^{18}F -FDS was solely related to enhanced blood pool effect, which is the manifestation of sepsis, similar uptake should have been observed in mice with *S. aureus* lung infection that were as sick as IPA mice. Further studies are needed to explain this finding.

8. While the autoradiography data is useful, the whole lung is lighting up. Can the authors show data where uninfected area within the same lung are included?

(Answer) The autoradiography images in the original manuscript have been replaced by images showing uninfected areas in the same lungs.

Supplementary Figure 7. (A) Autoradiographic image of ^{18}F -FDS and photograph of frozen *A. fumigatus*-infected lung sections. Mice with *A. fumigatus*-infected lungs were intravenously injected with 111 MBq of ^{18}F -FDS. The mice were sacrificed and their lungs

were removed, and 25 µm frozen sections of lung were cut using a cryostat. (B) Pathological examination of *A. fumigatus*-infected lung tissue.

9. In Figure S10, what tissue was used as the background?

(Answer) Background has been defined as the normal muscle of hind leg. The legend to this figure (now Figure S11) has been revised:

Supplementary Figure 11 Target (*A. fumigatus*-infected lung tissue)-to-background (normal muscle of hind leg) ratios of ¹⁸F-FDS (2 h) and ¹⁸F-FDG (1 h) post-injection (7.4 MBq).

Other points:

1. Due to changes in nomenclature, it would be better to use “Enterobacterales” instead of “Enterobacteriaceae”.

(Answer) We have replaced “Enterobacteriaceae” with “Enterobacterales” throughout the manuscript.

2. It would be important to cite other reports about FDS PET for infection imaging (PMID: 28848037, 27635025) including published clinical studies showing safety and efficacy (PMID: 27020679, 33853931)

(Answer) We have cited the recommended references related to infection imaging, including clinical studies showing safety and efficacy (PMID 28848037, 27635025, 27020679, and 33853931).

(Page 3, Lines 47–49) Because the sugar-free sweetener sorbitol is a metabolic substrate for Enterobacterales, the ability of 2-deoxy-2-¹⁸F-fluorosorbitol (¹⁸F-FDS) was assessed for its ability to image pathogenic *E. coli*²⁷, with this method found to be selective for Gram-negative pathogenic bacteria^{28, 29}.

(Page 7, Lines 154–155) ^{18}F -FDS PET could be utilized in patients receiving antifungal treatment to evaluate their microbial burden and to monitor the efficacy of treatment^{34,35}.

3. While ^{18}F -FDS has been evaluated for its ability to detect gliomas (Li et al. Mol. Imaging Biol. 2008), it was not found to be a good imaging agent for this indication. Please modify the introduction to clarify this, as it sounds misleading as written currently.

(Answer) We have modified this sentence and revised the “Introduction” section accordingly.

(Page 3, Lines 47–49) Because the sugar-free sweetener sorbitol is a metabolic substrate for Enterobacterales, the ability of 2-deoxy-2- ^{18}F -fluorosorbitol (^{18}F -FDS) was assessed for its ability to image pathogenic *E. coli*²⁷, with this method found to be selective for Gram-negative pathogenic bacteria^{28,29}.

4. Why is there no corresponding CT for panel b in Figure 3 and in Figure 6? This should be shown.

(Answer) We have replaced the previous image in panel b of Figure 3 with a new PET and CT merged image. We could not take CT images in Figure 6 because the scanner was unavailable at the time we monitored antifungal efficacy by ^{18}F -FDS PET. However, we could distinguish a region of *A. fumigatus*-infected myositis in Figure 6 based on Figure 2 and Supplementary Figure 2 (n = 41).

5. It is unclear (Figure 5) what the 24, 48 and 72 hours time-points mean.

(Answer) These three time points represent 24, 48, and 72 hours after *Aspergillus* brain infection. To avoid confusion, we have revised the legend to Figure 5d:

(Page 35, Lines 581–583) **d** Representative microPET/CT images of ^{18}F -FDS in normal and infected brains at 24 h (n = 5), 48 h (n = 5), and 72 h (n = 7; white arrows) post-infection (*p.i.*).

6. Table 1 and Table S1 seem to be identical. If so, please remove one of the tables.

(Answer) We have deleted Table S1.

Reviewer #2 (Remarks to the Author):

Innovative study evaluating *in vitro* and *in vivo* the new PET tracer fluorodeoxysorbitol for detecting invasive aspergillosis and specifically the differentiating between aspergillosis from other types of infection and cancer.

The study is well designed and clearly written.

Some major comments:

1. In the different *in vitro* and *in vivo* studies the uptake differences between fungal species, tumor model and *S. aureus* model are clearly described; the results of *E. coli* are mostly missing, and this Gram-negative bacterial type is of specific interest since it will also take up sorbitol.

(Answer) The “Introduction” and “Discussion” sections of the manuscript describe the value of ^{18}F -FDS in the detection of Gram-negative Enterobacterales such as *E. coli*. Several relevant publications have been cited. Because many preclinical and clinical studies have investigated the use of ^{18}F -FDS imaging in *E. coli* infection, the present study did not repeat the *in vivo* experiment using *E. coli*-infected mice. However, *E. coli* infection was used as a positive control in cell uptake assays.

2. The number of mice used in the different studies is unclear. If I'm correct then sometimes the number is only 3 or 4; not enough to draw significant conclusions.

(Answer) Except for autoradiography, we used a sufficient number of mice to draw significant conclusions. The number of animals used in each experiment is summarized in

the table:

Figure	Explanation		Number of animals (n)
Figure 2a	A. fumigatus -infected myositis model		9
Figure 2b	Correlation between ¹⁸ F-FDS and number of A. fumigatus cells		32
Figure 2c	> 3.1 Log (CFU/g)		20
	≤ 3.1 Log (CFU/g)		12
Figure 3d	¹⁸ F-FDS	A. fumigatus myositis and sterile inflammation model	4
		CT26 tumor model	6
		S. aureus myositis model	5
Figure 3e	¹⁸ F-FDG	A. fumigatus myositis and sterile inflammation model	5
		CT26 tumor model	6
		S. aureus myositis model	6
Figure 4c	¹⁸ F-FDS	Normal lung model	5
		A. fumigatus infected lung model	6
		S. aureus infected lung model	4
Figure 4c	¹⁸ F-FDG	Normal lung model	6
		A. fumigatus infected lung model	4
		S. aureus infected lung model	4
Figure 5f	¹⁸ F-FDS	A. fumigatus infected brain model (24 h p.i.)	5
		A. fumigatus infected brain model (48 h p.i.)	5
		A. fumigatus infected brain model (72 h p.i.)	7
	¹⁸ F-FDG	A. fumigatus infected brain model (72 h p.i.)	4
Figure 6c	Voriconazole	Before treatment	7
		8 days after treatment	7
		13 days after treatment	7
Figure 6d	Fluconazole	Before treatment	7
		8 days after treatment	5
		13 days after treatment	2
Table 1	Biodistribution		4
Supplementary Figure 2 and 3	A. fumigatus -infected myositis model		9
Supplementary Figure 6	¹⁸ F-FDS	A. fumigatus infected lung model	6
Supplementary Figure 7	¹⁸ F-FDS	Autoradiography of A. fumigatus infected lung model	2
Supplementary Figure 8	¹⁸ F-FDS	S. aureus infected lung model	4
Supplementary Figure 9	¹⁸ F-FDG		4
Supplementary Figure 10	¹⁸ F-FDG	A. fumigatus infected lung model	4
Supplementary Figure 12	¹⁸ F-FDS	B16F10 lung metastasis model	4
	¹⁸ F-FDG		4
Supplementary Figure 13	¹⁸ F-FDS	LPS induced lung inflammation model	5
	¹⁸ F-FDG		5
Supplementary Figure 15	¹⁸ F-FDS	A. fumigatus infected brain model (48 h p.i.)	5

3. The discussion should be rewritten; the first part (line 164-200) is more a repetition of the introduction. The start of the discussion should focus on the important findings of this study.

(Answer) As suggested by this reviewer, we removed the first paragraph of the “Discussion” (line 164–200) and replaced it with a summary of important findings (see below). Some information in the deleted paragraph was moved to the “Introduction” section.

(Page 9, Lines 166–171) ^{18}F -FDS showed selective concentration and prolonged retention in regions of *A. fumigatus* infection with low background activity, suggesting that this radiotracer could be used as a novel PET molecular imaging agent to obtain high quality images in the diagnosis of *A. fumigatus* infection. ^{18}F -FDS uptake differentiated *A. fumigatus* infection from other infectious and non-infectious causes. These imaging findings correlated well with the clinical course of IA, thus enabling the effects of treatment to be monitored.

4. In the discussion a part should be added how to implement this tracer in the clinics; especially since the authors also mention that there is only a narrow window for diagnosis. It will be difficult for most nuclear centers to produce F-sorbitol within a short time frame.

(Answer) ^{18}F -FDS can be synthesized rapidly from ^{18}F -FDG through simple reduction, which can be easily performed in most nuclear medicine facilities. Therefore, ^{18}F -FDS PET imaging will be clinically useful in the early diagnosis of IA. Even if ^{18}F -FDS cannot be prepared as quickly as needed, this tracer can enhance the accuracy of diagnosis. As mentioned in the “Introduction” and “Discussion” sections, precise diagnosis is just as important as early diagnosis for better outcome of IA. Empirical antifungal therapy, which is usually administered to patients with suspected IA, could be adjusted to a more definitive treatment based on the results of ^{18}F -FDS PET scanning. We have added appropriate

sentences to the “Discussion” section of the revised manuscript.

(Pages 9-10, Lines 189–195) One of the major advantages of ^{18}F -FDS as a tracer is that it can be synthesized rapidly from ^{18}F -FDG through simple reduction, which can be easily performed either on-site or at a nearby cyclotron. Therefore, ^{18}F -FDS PET imaging has potential application in the early diagnosis of IA in clinical settings. Even if ^{18}F -FDS cannot be prepared as quickly as needed, its high accuracy suggests that ^{18}F -FDS would be a promising tracer for more delayed diagnosis. Moreover, empirical antifungal therapy, which is usually administered to patients with suspected IA, could be adjusted to definitive therapy based on the results of ^{18}F -FDS PET scanning.

5. Parts of the introduction can be shortened.

(Answer) We have shortened the first paragraph of the “Introduction” and merged it with the second paragraph. Other parts of the Introduction have been revised.

6. The mentioned values of FDG uptake (0.892 - 0.372 etc) are very low; are these SUV calculations, ratios, MBq/cc. This should be further elaborated.

(Answer) The numbers 0.892 or 0.372 are ^{18}F -FDS uptake values, expressed as %ID/g, in the myositis model with different numbers of *A. fumigatus* conidia. These units have been added throughout the manuscript to reduce reader confusion.

(Page 5, Lines 96–99) ^{18}F -FDS uptake was significantly higher in infected than in normal muscle ($0.89 \pm 0.41\% \text{ID/g}$ vs. $0.20 \pm 0.11\% \text{ID/g}$, $P < 0.0001$). Below the cutoff level, however, ^{18}F -FDS uptake did not differ significantly in infected and normal muscle ($0.37 \pm 0.18\% \text{ID/g}$ vs. $0.26 \pm 0.07\% \text{ID/g}$, $P = 0.0531$) (Fig. 2c).

7. The myositis model: is that really an infection? Myositis is an inflammatory disease; the authors injected *A.fumigatus* in the right shoulder; then this is not myositis but a fungal infection focus in the soft tissue of the right shoulder

(Answer) Myositis is usually defined as inflammation of the muscle. Infectious myositis is the usual term to describe muscle infection by bacteria, fungi, or viruses⁶, a term used in other imaging studies in mice^{7,8}. Because we also analyzed mice with sterile inflammatory myositis induced by heat killed *A. fumigatus* conidia, we corrected “myositis” to “*A. fumigatus*-infected myositis” or “*S. aureus*-infected myositis” in this manuscript.

8. Limitations are missing in the discussion

(Answer) We have expanded our discussion of study limitations in the Discussion section of the revised manuscript, as described below:

(Pages 11, Lines 216–229) This study had several limitations. Because ¹⁸F-FDS also accumulates in Enterobacterales²⁷, ¹⁸F-FDS PET imaging could not differentiate *A. fumigatus* infection from Enterobacterales infection. Clinically, however, IA is mainly diagnosed in immunocompromised hosts and generally manifests as lung, sinus, and CNS infections, whereas Enterobacterales mainly cause intra-abdominal infection regardless of host immune status. The combination of ¹⁸F-FDS PET and appropriate clinical information may have a role in the differential diagnosis and localization of IA. Testing also showed that ¹⁸F-FDS was taken up *in vitro* by cells of other fungi, including *R. arrhizus* and *C. albicans*. ¹⁸F-FDS PET imaging of these fungi has not been evaluated *in vivo* to date. In contrast to mold infection, candida is an infrequent cause of pulmonary, brain, and sinus infections, and is relatively easy to diagnose microbiologically. Infection with *Rhizopus*, one of the most frequent causes of invasive mucormycosis, is usually fatal and may be difficult to distinguish from IA. Mucormycosis is increasingly reported related to COVID-19. Although

most of these patients had rhino-orbital cerebral disease, some also had pulmonary disease⁴². Additional studies are needed to evaluate the usefulness of ¹⁸F-FDS PET imaging in the diagnosis of *Rhizopus* infection.

Reviewer #3 (Remarks to the Author):

This spectacular paper reports on *in vivo* imaging of invasive aspergillosis using 2-deoxy-2-¹⁸F-fluorosorbitol (¹⁸F-FDS) PET, which clearly visualized *A. fumigatus* infection of the lungs, brain, and muscles in mouse models. Even more spectacular, this PET imaging did also show promise in assessing treatment success, opening doors for new imaging technologies for treatment monitoring in these often difficult to treat infections.

Comments:

1. In the introduction in the section for COVID associated aspergillosis it would be good to mention the Guidance on classification and treatment published recently in Lancet ID by Koehler et al [https://doi.org/10.1016/S1473-3099\(20\)30847-1](https://doi.org/10.1016/S1473-3099(20)30847-1)

(Answer) We have revised these sentences and cited this paper in the “Introduction” section.

(Page 2, Lines 23–25) Several recent reports have described COVID-19-associated pulmonary aspergillosis (CAPA)^{9, 10, 11, 12}, raising concerns that CAPA could worsen outcomes of COVID-19¹³.

(Pages 2-3, Lines 38–39) In patients with mixed infection such as CAPA, the lack of specific radiologic signs of IPA complicates its radiologic diagnosis¹³.

2. An important limitation was that the ¹⁸F-FDS uptake by *A. fumigatus*, *R. arrhizus*, and *C. albicans* was comparable to its uptake by *E. coli*, so ¹⁸F-FDS may not be useful for differentiating different pulmonary mold infections (aspergillosis versus Mucormycosis) and may also not differentiate mold infection from gram negative infection. Please discuss more extensively, also considering the emergence of COVID associated Mucormycosis. See

Preprint here with 80 cases including many pulmonary cases
https://papers.ssrn.com/sol3/papers.cfm?abstract_id=3844587.

(Answer) This limitation is discussed more extensively in the “Discussion” section of the revised manuscript.

(Pages 11, Lines 216–229) This study had several limitations. Because ^{18}F -FDS also accumulates in Enterobacterales²⁷, ^{18}F -FDS PET imaging could not differentiate *A. fumigatus* infection from Enterobacterales infection. Clinically, however, IA is mainly diagnosed in immunocompromised hosts and generally manifests as lung, sinus, and CNS infections, whereas Enterobacterales mainly cause intra-abdominal infection regardless of host immune status. The combination of ^{18}F -FDS PET and appropriate clinical information may have a role in the differential diagnosis and localization of IA. Testing also showed that ^{18}F -FDS was taken up *in vitro* by cells of other fungi, including *R. arrhizus* and *C. albicans*. ^{18}F -FDS PET imaging of these fungi has not been evaluated *in vivo* to date. In contrast to mold infection, candida is an infrequent cause of pulmonary, brain, and sinus infections, and is relatively easy to diagnose microbiologically. Infection with *Rhizopus*, one of the most frequent causes of invasive mucormycosis, is usually fatal and may be difficult to distinguish from IA. Mucormycosis is increasingly reported related to COVID-19. Although most of these patients had rhino-orbital cerebral disease, some also had pulmonary disease⁴². Additional studies are needed to evaluate the usefulness of ^{18}F -FDS PET imaging in the diagnosis of *Rhizopus* infection.

3. The discussion starts very broadly. Its fine to do it this way but would consider to add a paragraph first that shortly summarizes the most important findings of your study specifically.

(Answer) We have removed the first paragraph of the “Discussion” section (line 164–200) because it overlapped with the “Introduction”, and replaced it with a summary of important

findings. Some information in the deleted paragraph was moved to the “Introduction” section and other sentences were revised.

References

1. Díaz, L. G., López-Corral, L., Chinchilla, L. M., Caballero, D. & Tamayo, P. Mucormycosis with pulmonary, renal, pancreatic and spleen involvement detected by ^{18}F -FDG PET/CT in a patient who underwent allogeneic stem cell transplantation. Report of case. *Trends in Transplantation* **12**, 1-3 (2019).
2. Sharma, P., Mukherjee, A., Karunanithi, S., Bal, C. & Kumar, R. Potential role of ^{18}F -FDG PET/CT in patients with fungal infections. *AJR Am. J. Roentgenol.* **203**, 180-189 (2014).
3. Hod, N. et al. Detection of disseminated aspergillosis on FDG PET/CT in a patient with acute lymphoblastic leukemia. *Isr. Med. Assoc. J.* **20**, 717-719 (2018).
4. Lazarescu, R. E. & Vinelli, M. Dissemination of invasive aspergillosis: diagnostic and management dilemmas. *BMJ Case Rep.* **2014**, bcr2014204642 (2014).
5. Ordonez, A. A. et al. Imaging Enterobacterales infections in patients using pathogen-specific positron emission tomography. *Sci. Transl. Med.* **13**, eabe9805 (2021).
6. Crum-Cianflone, N.F. Bacterial, fungal, parasitic, and viral myositis. *Clin. Microbiol. Rev.* **21**, 473-494 (2008).
7. Yao, S. et al. Infection imaging with ^{18}F -FDS and first-in-human evaluation. *Nucl. Med. Biol.* **43**, 206-214 (2016).
8. Weinstein, E. A. et al. Imaging Enterobacteriaceae infection in vivo with ^{18}F -fluorodeoxysorbitol positron emission tomography. *Sci. Transl. Med.* **6**, 259ra146 (2014).

Reviewers' Comments:

Reviewer #1:

Remarks to the Author:

This is a revised manuscript by Kim et al evaluating the noninvasive detection of *Aspergillus* infections using 2-deoxy-2-¹⁸F-fluorosorbitol (¹⁸F-FDS) positron emission tomography (PET).

1. The revised in vitro uptake (Bq/μg; Supplementary Fig. 1) no longer shows increased uptake by fungi and is similar to *S. aureus*, which has been used a negative control for this study. This is critical as uptake per mass of the organism should be much higher to be able to visualize the infection over the background host tissues.

2. Why are the revised ¹⁸F-FDS biodistribution values in mice with *A. fumigatus*-infected myositis different from the ones presented earlier?

3. I remain concerned about the high ¹⁸F-FDS PET background in the *Aspergillus*-infected animals, which is not noted in the controls (Figure 4).

4. While dynamic PET was performed, I could not find those data.

Reviewer #2:

Remarks to the Author:

Thank you, happy with the provided answers. Congratulations with this nice study.

Reviewer #1 (Remarks to the Author):

This is a revised manuscript by Kim et al evaluating the noninvasive detection of *Aspergillus* infections using 2-deoxy-2-¹⁸F-fluorosorbitol (¹⁸F-FDS) positron emission tomography (PET).

1. The revised *in vitro* uptake (Bq/μg; Supplementary Fig. 1) no longer shows increased uptake by fungi and is similar to *S. aureus*, which has been used a negative control for this study. This is critical as uptake per mass of the organism should be much higher to be able to visualize the infection over the background host tissues.

(Answer) Thank you for your helpful comments. As you pointed out, when the *in vitro* ¹⁸F-FDS uptake of microbes was normalized to protein, a significant reduction of uptake was observed in the filamentous fungi in comparison with *E. coli*, a positive control in our study (Fig. 1 of submitted manuscript and Supplementary Fig. 1). As noted by Lai et al.¹, who recently carried out an ¹⁸F-FDS uptake assay in *Aspergillus* spp., direct comparisons of fungal burden with bacterial burden are very difficult because of differences in growth patterns, dynamics, and metabolism. For the *in vitro* uptake assays, we counted and adjusted the number of colony forming units (CFUs) or conidia to 1×10^7 . In addition, the amount of protein for each pathogen was measured to normalize the biomass. Because conidia (diameter of 5–18 μm for *Rhizopus* spp. and 2–3 μm for *A. fumigatus*) are much larger than bacterial cells (cylindrical diameter of 1–2 μm, with a radius of approximately 0.5 μm for *E. coli*), the amount of protein in fungi was much higher for the same number of cells. The degree of increase in the amount of protein was also very different between the fungi and bacteria when measured after 2 h of ¹⁸F-FDS incubation (R Fig. 1 of response letter). In media, especially broth media, not all conidia are activated and germinate to form hyphae, even after relatively long incubation periods. Some conidia remain dormant throughout *in vitro* culture, and it is probable that these conidia may only passively take

up very small amounts of a carbon source such as sorbitol. The conidial germination rate of *A. fumigatus in vitro* is also affected by different culture and environmental conditions². Therefore, when FDS uptake is normalized by the total number of conidia or amount of protein, as long as a substantial percentage of conidia remain dormant, the extent of uptake is likely to be relatively low. To explain this, we measured the conidial germination rate of *A. fumigatus* in different growth media with or without sorbitol. We observed that under various culture conditions, the germination rate was 0–40% at 6 h of culture, reaching a maximum of 80% at 10 h of culture (R Fig. 2 of response letter). Furthermore, we also visualized the mixed-growth stage of *A. fumigatus* in broth media at specific time points (20 and 48 h) using fluorescent sorbitol (sorbital-ZW800-1)³. The near-infrared fluorescence imaging rather obviously showed that more sorbitol was accumulated in germinated *A. fumigatus* than in conidia (Supplementary Fig. 2). Germination is an essential phase of invasive *Aspergillus* infection in living subjects⁴. In most infected tissue, *A. fumigatus* is observed as well-developed hyphae. We have also presented histological findings showing this in our manuscript (Figs. 4 and 5 of manuscript and Supplementary Fig. 5). The hyphal stage appears to exhibit more active uptake of sorbitol than conidia (Supplementary Fig. 2). Indeed, Lai et al.¹ also reported similar findings using ³H-sorbitol uptake assays. Therefore, in order to use the results of the *in vitro* ¹⁸F-FDS uptake assays to predict feasibility in *A. fumigatus* infection imaging, it is necessary to quantify the fungal biomass with the exclusion of dormant conidia *in vitro*, which is quite difficult.

In summary, to measure the microbiological biomass for normalization of *in vitro* uptake assay, it is necessary to consider the *in vitro* growth characteristics of filamentous fungi, which are very different to those of bacteria. Despite these limitations, we observed significantly higher uptake of ¹⁸F-FDS in *A. fumigatus* than in *S. aureus* (negative control), regardless of the method of normalization (protein or number of conidia), although this uptake was much less than that observed in *E. coli* (positive control). We finally presented the *in vitro* ¹⁸F-FDS uptake assay with a reduced number of bacteria and conidia (from 10⁷ to 10⁶ CFU). With this method, the *in vitro*

uptake of ^{18}F -FDS was increased in *A. fumigatus* (Supplementary Fig. 1).

Taken together, as we observed clear visualization of *A. fumigatus*-infected lesions *in vivo* (muscle, brain, and lung), and also differentiated *A. fumigatus* infection from sterile inflammation and *S. aureus* infection, we speculate that the relatively low *in vitro* ^{18}F -FDS uptake when normalized according to the amount of protein appeared to be due to the differences in *A. fumigatus* growth between biotic and abiotic environments. Therefore, we consider that it is worth continuing the evaluation of ^{18}F -FDS as a radiotracer for diagnosis of invasive aspergillosis in preclinical and clinical practice.

R Figure 1. Comparison of protein amount between the bacteria (1 mL of LB media concentrated to 1×10^7 colony forming units [CFU]/mL) and fungi (1 mL of YPD media concentrated to 1×10^7 conidia or CFU /mL) used for the *in vitro* ^{18}F -FDS uptake assay. The amount of protein was measured at the beginning of the *in vitro*-uptake assay (10 min), and after 30, 60, and 120 min of ^{18}F -FDS incubation.

R Figure 2. Conidial germination in different culture media of potato dextrose broth (PDB), yeast extract sucrose (YES), yeast extract peptone dextrose broth (YPD), distilled water (dH₂O), and phosphate-buffered saline (PBS) with 0.1% Tween 20 with (0.1% and 1%) or without sorbitol. To measure the germination rate, conidia were observed in microtitration plates by differential interference contrast (DIC) microscopy every 2 h. At each time point, 100 conidia were counted and the percentage of germination in each medium was estimated in triplicate. a. Representative microscopic image of conidial germination in YES media according to time point. b. The rate of germination under different culture conditions. At all tested temperatures and media, no germ tubes were formed within 4 h, but they began to form after 6 h. The germination rate of conidia showed significant differences according to the culture conditions. The germination was slightly faster in YES medium than in PDB broth medium. Sorbitol addition enhanced the germination in PDB, but not in YES. Over 24 h, germinated conidia were not observed in distilled water or PBS with 0.1% Tween 20 in the presence or absence of sorbitol.

We have replaced Supplementary Figure 1 and added Supplementary Figure 2. Furthermore, we have added appropriate text to the “Results”, “Discussion”, and “Methods” sections of the revised manuscript.

Supplementary Figure 1. Cellular uptake of ^{18}F -FDS by *A. fumigatus*, *R. arrhizus*, *C. albicans*, *S. aureus* (-), and *E. coli* (+) cells 1 h after treatment with ^{18}F -FDS. Data are expressed as the absolute accumulation activity (Bq) \pm SD (normalized by protein of 1×10^6 cells) of four replicate experiments ($^*P < 0.05$).

Supplementary Figure 2. Confocal microscopy demonstrates live binding of sorbitol-ZW800-1 to different growth stages of *A. fumigatus*. *A. fumigatus* was cultured on potato dextrose agar plates for 5–7 days at 30°C and colonies were harvested using saline containing 0.1% (v/v) Tween 20. The suspensions were vortexed to release the conidia and filtered through a 40 µm cell strainer to remove clumps and hyphae. Conidia were counted using a hemocytometer, and 1×10^7 *A. fumigatus* conidia and 50 µM of sorbitol-ZW800-1 were co-cultured for up to 48 h in 1 mL of YPD broth in 24-well plates at 37°C without agitation. Co-cultured *A. fumigatus* were observed at 20 and 48 h using laser scanning confocal microscopy (LSM 800, ZEISS, Germany). Before microscopic examination, cultured *A. fumigatus* were washed twice, re-suspended in PBS, and transferred to a confocal dish. Fluorescence of ZW800-1 (red) was obtained using laser filter excitation at 561 nm and emission at 650–750 nm. Image acquisition and analysis were performed using ZEN 2.6, blue edition (ZEISS, Germany). Merged differential interference contrast (DIC) and laser scanning images show more sorbitol accumulated in germinated *A. fumigatus* than in conidia. (A) Twenty hours of incubation \times 200; (B) forty-eight hours incubation \times 400.

(Page 4, Lines 108) We also tried to visualize *A. fumigatus* directly under a confocal fluorescence microscopy using near-infrared fluorescent sorbitol (sorbitol-ZW800-1) and showed that sorbitol-ZW800-1 mainly accumulated in germinated *A. fumigatus* (Supplementary Fig. 2).

(Page 11, Lines 264) An *in vitro* uptake assay demonstrated that ^{18}F -FDS uptake in *A. fumigatus* was as high as in *E. coli* when it was normalized to 1×10^7 conidia or CFU. However, when *in vitro* ^{18}F -FDS uptake of microbes was normalized to protein, significant reduction of uptake was observed in fungi in comparison with *E. coli* (Supplementary Fig. 1). As fungi and bacteria have very different growth patterns, dynamics, and metabolism, determination of a universal indicator to normalize the biomass for both is difficult⁴². In broth media, not all conidia of *A. fumigatus* are activated and germinated to form hyphae, and some remain as dormant conidia, even after relatively long incubation periods⁴³. As hyphae take up more sorbitol than conidia⁴², if dormant conidia remain in the culture media, the uptake value might be underestimated when normalized with protein. In a further experiment with near infrared fluorescence microscopy, we were able to directly observe accumulation of fluorescent sorbitol (sorbitol-ZW800-1) in germinated *A. fumigatus* in comparison with conidia (Supplementary Fig. 2). In most *in vivo* infected tissue, *A. fumigatus* is observed as well-developed hyphae, as germination is an essential phase in invasive Aspergillus infection⁴, and we also observed well-developed hyphae in our histological findings (Figs. 4 and 5 of the manuscript and Supplementary Fig. 5). We subsequently observed significantly higher ^{18}F -FDS uptake in the infected tissue than in the background tissue on PET imaging. Therefore, we speculate that because of differences in the growth of *A. fumigatus* between biotic and abiotic environments, *in vitro* uptake might be underestimated when normalized according to the amount of protein.

(Page 16, Lines 361) Confocal Microscopy *A. fumigatus* was cultured on potato dextrose agar plates for 5–7 days at 30°C, and colonies were harvested using PBS containing 0.1% (v/v) Tween 20. The

suspensions were vortexed to release the conidia and filtered through a 40 µm cell strainer to remove clumps and hyphae. Conidia were counted using a hemocytometer, and 1×10^6 *A. fumigatus* conidia and 50 µM of sorbitol-ZW800-1 were co-cultured for up to 48 h in 1 mL of YPD broth in 24-well plates at 37°C without agitation. Sorbitol-ZW800-1 was prepared as described previously⁴⁶. Co-cultured *A. fumigatus* were observed at 20 and 48 hours using laser scanning confocal microscopy (LSM 800, ZEISS, Germany). Before microscopic examination, cultured *A. fumigatus* were washed twice, re-suspended in PBS, and transferred to a confocal dish. Fluorescence of ZW800-1 (red) was obtained using laser filter excitation at 561 nm and emission at 650–750 nm. Image acquisition and analysis were performed using ZEN 2.6, blue edition (ZEISS, Germany).

2. Why are the revised ^{18}F -FDS biodistribution values in mice with *A. fumigatus*-infected myositis different from the ones presented earlier?

(Answer) To ensure accuracy and reproducibility, the biodistribution study was performed again using more mice with *A. fumigatus*-infected myositis (n = 8, each). We therefore replaced Table 1 and revised the results and the numbers of mice in the manuscript as follows.

(Page 33, Lines 553) Table 1. Biodistribution of ^{18}F -FDS (2 h, 7.4 MBq, n = 8) in mice with *A. fumigatus*-infected myositis.

Organs	%ID/g (2 h)
Blood	0.22 ± 0.05
Heart	0.82 ± 0.14
Lung	0.97 ± 0.16
Liver	0.46 ± 0.15
Spleen	1.59 ± 0.25
Stomach	0.74 ± 0.42
Intestine	1.75 ± 0.93
Kidney	0.79 ± 0.17
Pancreas	1.42 ± 0.26
Bone	0.72 ± 0.13
Brain	0.33 ± 0.04
Skin	0.51 ± 0.12

Normal muscle	0.28 ± 0.03
Infected muscle	1.84 ± 0.38

(Page 6, Lines 156) The *in vivo* biodistribution study performed 2 h after tracer injection also showed selective concentration of ¹⁸F-FDS in the shoulders with *A. fumigatus*-infected myositis, with concentrations of ¹⁸F-FDS in these shoulders being 8.4-, 1.8-, 4.0-, 5.5-, and 6.6-fold higher than in blood, lungs, liver, brain, and normal muscle, respectively (Table 1).

3. I remain concerned about the high ^{18}F -FDS PET background in the *Aspergillus*-infected animals, which is not noted in the controls (Figure 4).

(Answer) Thank you for this important comment. We have also questioned this throughout our research, and we mentioned the blood pool effect in our previous answers and the revised manuscript. In the most recent study on ^{18}F -FDS PET imaging in *Aspergillus* spp., carried out by Lai et al., the dynamic PET/CT imaging showed that the activity of ^{18}F -FDS in the blood appeared to be maintained longer in *A. fumigatus*-infected mice than in healthy controls, which might be due to a hemodynamic change, such as systemic vasodilation caused by serious infection¹. A blood pooling effect of ^{18}F -FDS was also observed on PET scans of *Enterobacteriales*-infected patients in the clinical study by Ordonez et al.⁵. In our study, we did not observe a significant ^{18}F -FDS background in *A. fumigatus* muscle (Fig. 2, Fig. 3a, Fig. 6, and Supplementary Fig. 3) or a brain infection mouse model (Fig. 5 and Supplementary Fig. 16), which might be because these infections were not as severe as the pulmonary infection and did not cause the same hemodynamic changes.

However, if the *in vivo* background activity of ^{18}F -FDS was solely due to a blood pooling effect, we should have observed a similar finding in *S. aureus* lung infection. These mice were as sick as those with pulmonary aspergillosis, but we did not observe significant background activity of ^{18}F -FDS in *S. aureus* lung infection (Fig. 4 and Supplementary Fig. 9). As systemic aspergillosis followed by pulmonary infection is not uncommon in immunocompromised hosts, it might be possible that seeding of *A. fumigatus* in other organs contributed to increased background activity of ^{18}F -FDS. Therefore, we tested whether or not *A. fumigatus* could be cultured from other organs (liver, kidneys) of pulmonary infection mice. For this experiment, mice infected with *A. fumigatus* in the lungs were sacrificed 48 h after infection (same mouse model used for the ^{18}F -FDS PET imaging) and their liver and kidneys were extracted, weighed, and homogenized in 5 mL of sterile saline. The homogenates were serially diluted, spread on

potato dextrose agar plates, and incubated at 30°C for 48 h. The results revealed that *A. fumigatus* could be cultured from the liver and kidneys of some mice infected with *A. fumigatus* in the lungs (R Table 1). This result, along with the blood pooling effect, might partially explain the increased background activity of ¹⁸F-FDS in mice with *A. fumigatus* pulmonary infection. However, when Ordonez et al.⁵ scanned *Enterobacteriales*-infected patients with ¹⁸F-FDS PET they observed increased ¹⁸F-FDS activity in uninfected body fluids (such as the fluid of large joints, pleural, and peritoneal effusions) that were distant from infected lesion, and further studies are needed to provide a clear explanation of the increased activity of ¹⁸F-FDS in areas not directly subject to infection.

Although the background activity of ¹⁸F-FDS in *A. fumigatus*-infected mice increased to an extent, we still observed a sufficient target-to-background ratio (infected lung to muscle), which was not observed on ¹⁸F-FDG imaging (0.80 ± 0.07 , Supplementary Fig. 12). In addition, in muscle and brain infection, ¹⁸F-FDS clearly visualized the area of infection. Therefore, the potential of ¹⁸F-FDS as a radiotracer for the diagnosis of invasive aspergillosis needs to be continuously evaluated in preclinical and clinical practice.

R Table 1. Isolation of *A. fumigatus* in liver and kidney tissues in *A. fumigatus* lung-infection mice.

	Log ₁₀ CFU/g of kidneys	Log ₁₀ CFU/g of liver
mice 1	2.9	0.0
mice 2	3.2	2.5
mice 3	3.2	0.0
mice 4	2.9	0.0
mice 5	0.0	0.0
mice 6	2.9	0.0
mice 7	3.2	2.7

Mean	2.6	0.7
Median	2.9	0.0
Standard deviation	1.07	1.2

We have added new text to the “Discussion” section as follows.

(Page 10, Lines 253) In the most recent study on ^{18}F -FDS uptake in *Aspergillus* spp., dynamic PET/CT imaging showed that the radioactivity of ^{18}F -FDS in the blood appeared to be maintained longer in *A. fumigatus* infected mice than in healthy control mice, which might be due to hemodynamic change, such as systemic vasodilation caused by serious infection⁴².

(Page 11, Lines 260) Even if pathogen-specific uptake and an increased blood pool effect coexist in IPA mice, the potential of ^{18}F -FDS PET imaging should not be overlooked, given that in diverse IA models the target-to-background ratio is high in brain and muscle, as well as in lung (4.62 ± 1.15), which is in contrast to ^{18}F -FDG PET imaging (0.80 ± 0.07 , Supplementary Fig. 12).

4. While dynamic PET was performed, I could not find those data.

(Answer) We have attached a dynamic image as an MP4 file. The name of this movie file is “FDS dynamic for 2 h_mp4”.

Reviewer #2 (Remarks to the Author):

Thank you, happy with the provided answers. Congratulations with this nice study.

(Answer) I deeply appreciate both the criticism and appreciation you have shown in our paper. Thank you very much.

References

1. Lai, J. et al. Evaluation of 2-[¹⁸F]-fluorodeoxysorbitol PET imaging in preclinical models of aspergillus infection. *J. Fungi (Basel)* **8**, 25 (2021).
2. Meletiadis, J., Meis, J. F., Mouton, J. W. & Verweij, P. E. Analysis of growth characteristics of filamentous fungi in different nutrient media. *J. Clin. Microbiol.* **39**, 478-484 (2001).
3. Lee, S. et al. Near-infrared fluorescent sorbitol probe for targeted photothermal cancer therapy. *Cancers (Basel)* **11**, 1286 (2019).
4. Dagenais, T. R. & Keller, N. P. Pathogenesis of *Aspergillus fumigatus* in invasive aspergillosis. *Clin. Microbiol. Rev.* **22**, 447-465 (2009).
5. Ordonez, A. A. et al. Imaging Enterobacterales infections in patients using pathogen-specific positron emission tomography. *Sci. Transl. Med.* **13**, eabe9805 (2021).